# Nucleosomes influence multiple steps during replication initiation

**Ishara F Azmi[1], Shinya Watanabe[2], Michael F Maloney[1], Sukhyun Kang[1,3], Jason A Belsky[4,5], David M MacAlpine[4], Craig L Peterson[2], Stephen P Bell[1]***

[1]Department of Biology, Howard Hughes Medical Institute, Massachusetts Institute of Technology, Cambridge, United States; [2]Program in Molecular Medicine, University of Massachusetts Medical School, Worcester, United States; [3]Center for Genomic Integrity, Institute for Basic Science, Ulsan, South Korea; [4]Department of Pharmacology and Cancer Biology, Duke University Medical Center, Durham, United States; [5]Program in Computational Biology and Bioinformatics, Duke University, Durham, United States

**Abstract** Eukaryotic replication origin licensing, activation and timing are influenced by chromatin but a mechanistic understanding is lacking. Using reconstituted nucleosomal DNA replication assays, we assessed the impact of nucleosomes on replication initiation. To generate distinct nucleosomal landscapes, different chromatin-remodeling enzymes (CREs) were used to remodel nucleosomes on origin-DNA templates. Nucleosomal organization influenced two steps of replication initiation: origin licensing and helicase activation. Origin licensing assays showed that local nucleosome positioning enhanced origin specificity and modulated helicase loading by influencing ORC DNA binding. Interestingly, SWI/SNF- and RSC-remodeled nucleosomes were permissive for origin licensing but showed reduced helicase activation. Specific CREs rescued replication of these templates if added prior to helicase activation, indicating a permissive chromatin state must be established during origin licensing to allow efficient origin activation. Our studies show nucleosomes directly modulate origin licensing and activation through distinct mechanisms and provide insights into the regulation of replication initiation by chromatin.

*For correspondence: spbell@mit. edu

## Introduction

The eukaryotic genome is packaged into a condensed form known as chromatin that presents a barrier to DNA-associated processes. Chromatin is primarily composed of nucleosomes, each of which consists of ~147 base pairs of DNA wrapped around a histone octamer. The location and modification state of nucleosomes is dynamic, regulates access to the DNA and partitions the genome into distinct chromatin states (*Clapier and Cairns, 2009*). Nucleosome positioning and modifications influence all DNA processes including replication, transcription, repair and recombination. Thus, maintaining appropriate chromatin states across the genome is critical for cellular viability (*Hargreaves and Crabtree, 2011*). Although there is a growing wealth of knowledge concerning the impact of nucleosomes on gene expression, significantly less is known about the role of nucleosomes in regulating DNA replication.

Proper eukaryotic DNA replication requires the temporal separation of two key events: origin licensing and origin activation (*Li and Araki, 2013*; *Siddiqui et al., 2013*). During G1, origin licensing is initiated by origin-recognition complex (ORC) binding to replication origin DNA. ORC then recruits Cdc6 and Cdt1 and these proteins load two inactive Mcm2-7 replicative DNA helicases around the origin DNA (*Bell and Labib, 2016*). Origin activation is temporally separated from origin licensing and occurs during S phase. S-phase cyclin-dependent kinases and the Dbf4-dependent

**eLife digest** Each human cell contains more than two meters of DNA. To fit this length into a cell, remodeling enzymes compact the DNA by helping it to bind to specific proteins. This compaction has the side effect of making the DNA harder to access.

DNA replication is one process that requires access to the DNA. Replication occurs each time a cell divides, so that each newly formed cell receives a full set of genetic material. DNA replication starts simultaneously at hundreds of sites across the DNA. At each of these sites, cells assemble a protein called a replicative helicase. Helicases play a important role in many steps of DNA replication, but their most fundamental role is to separate the two DNA strands that make up the double helix; these strands then act as templates during replication.

A helicase is initially inactive when loaded at a replication start site. Additional proteins then bind to the helicase to activate it. Studies have shown that DNA compaction influences DNA replication, but it was not known exactly how compacted DNA affects helicase loading and activation.

To investigate the effects of compacted DNA during replication in more detail, Azmi et al. created different types of compacted DNA molecules using various remodeling enzymes. Some of the compacted DNAs directly prevented the binding of a protein that is required to load the helicase to the replication start site. In addition, the compaction reduced the number of sites on the DNA where replication could begin. Other types of compacted DNA allowed the helicase to be loaded normally, but inhibited the subsequent activation of the helicase. However, treating these DNA types with particular remodeling enzymes restored helicase activation to normal levels.

Overall, the findings presented by Azmi et al. suggest that cells can control helicase loading and activation independently by compacting DNA in different ways. Such control is important to ensure that each time a cell divides, it fully replicates its entire DNA.

Cdc7 kinase (DDK) drive recruitment of two helicase-activating proteins, Cdc45 and GINS, forming the active replicative helicase, the Cdc45/Mcm2-7/GINS (CMG) complex (*Ilves et al., 2010*; *Tanaka and Araki, 2013*). Recruitment of DNA polymerases and their accessory proteins to the CMG complex forms a bidirectional pair of replisomes. The majority of these events have been reconstituted in vitro using purified proteins and naked DNA (*Yeeles et al., 2015*, *2017*).

Chromatin is proposed to influence multiple aspects of replication initiation including origin licensing, origin activation and the time of replication initiation within S phase. Origin DNA is nucleo-some-free to allow ORC DNA binding and, once bound, ORC positions origin-proximal nucleosomes (*Berbenetz et al., 2010*; *Eaton et al., 2010*). Repositioning of origin-proximal nucleosomes reduces helicase loading and origin function (*Lipford and Bell, 2001*; *Simpson, 1990*), and loaded helicases appear to interact with these nucleosomes (*Belsky et al., 2015*). Local chromatin states have also been implicated in the activation of eukaryotic origins, each of which is predisposed to initiate earlier or later in S phase (*Rhind and Gilbert, 2013*). How chromatin modulates these events and whether specific chromatin regulators impact replication initiation events is unclear.

Chromatin-remodeling enzymes (CREs) play a major role in determining the chromatin landscape across the genome (*Struhl and Segal, 2013*). CREs are multi-protein complexes that use the energy of ATP binding and hydrolysis to assemble, move, slide or alter the composition of nucleosomes (*Clapier and Cairns, 2009*; *Papamichos-Chronakis and Peterson, 2012*). Four subfamilies of CREs are conserved from yeast to humans: ISWI, SWI/SNF, INO80, and CHD. Members of the ISWI and CHD subfamilies typically function in nucleosome assembly, and they can create regularly spaced nucleosomal arrays by ATP-dependent sliding of nucleosomes (*Hamiche et al., 1999*; *Längst et al., 1999*). Similarly, members of the SWI/SNF subfamily mobilize nucleosomes in cis, but these enzymes can also evict nucleosomal histones or eject entire nucleosomes (*Clapier et al., 2016*). Consequently, these CREs typically promote enhanced accessibility of nucleosomal DNA. Finally, members of the INO80 subfamily conduct the post-replicative removal of a particular histone within a nucleosome, and sequential replacement with either a canonical or a variant histone, a process termed nucleosome editing (*Mizuguchi et al., 2004*; *Papamichos-Chronakis et al., 2011*). Notably, some members of the INO80 subfamily can also catalyze nucleosome sliding (e.g. yeast INO80-C)

(*Shen et al., 2003*). Although different CREs can exert a differential impact on nucleosomes, the current view is that each of these enzymes use ATP-dependent DNA translocation as a central mechanism for their activities.

Various CREs have been implicated in the regulation of DNA replication (*MacAlpine and Almouzni, 2013*). For instance, ISW1-containing remodeling complexes interact with replisome proteins (*Poot et al., 2005*) and Chd1 negatively regulate replication initiation (*Biswas et al., 2008*). Similarly, SWI/SNF stimulates replication initiation at specific yeast origins (*Flanagan and Peterson, 1999*) and is associated with a subset of human origins (*Euskirchen et al., 2011*).

Although elimination of different CREs influences DNA replication, whether these effects are direct or indirect and the specific events of replication that are impacted remain elusive. CREs impact multiple processes including transcription, histone modification, and nucleosome assembly (*Clapier and Cairns, 2009*) leaving open the possibility of indirect effects. In addition, cells express multiple members of each CRE family and overlapping functions of these enzymes could mask the effects of single CRE deletions (*Tsukiyama et al., 1999*). Although the simultaneous deletion of multiple CREs could overcome this issue, in many cases these are lethal events (*Monahan et al., 2008*; *Tsukiyama et al., 1999*).

Here we describe origin-dependent in vitro replication assays using nucleosomal DNA templates. To address how different nucleosomal states impact DNA replication, we investigated nucleosomal templates that were remodeled by different CREs. Consistent with in vivo studies, these templates showed distinct replication capacities. Most of the nucleosomal DNA templates permitted origin licensing, but ISW2- and Chd1-remodeled templates reduced the efficiency of this event by positioning nucleosomes over the origin DNA, decreasing ORC DNA binding and helicase loading. Although permissive for origin licensing, SWI/SNF- and RSC-remodeled templates showed reduced CMG formation and origin activation. Addition of specific CREs improved replication initiation from these templates but only if the CRE was added prior to CMG formation. Our findings show that local nucleosome status differentially modulates two steps during replication initiation and that specific CREs establish permissive and restrictive states for replication initiation.

## Results

### Reconstitution of Mcm2-7 helicase loading using nucleosomal DNA

To investigate the impact of chromatin on replication initiation, we first reconstituted origin licensing using nucleosomal DNA templates. To this end, we used purified ISW1a, Nap1 and budding yeast histone octamers (*Figure 1—figure supplement 1A and B*) to assemble nucleosomes on a 3.8 kb linear fragment of *Saccharomyces cerevisiae* DNA that included the *ARS1* replication origin (*Mizuguchi et al., 2012*). We optimized the ratio of DNA to histone octamers to assemble regularly-spaced nucleosome arrays (*Figure 1A* and *Figure 1—figure supplement 2A*). After nucleosomes were remodeled, ISW1a, Nap1 and free histones were removed from the template (*Figure 1—figure supplement 2B*) to provide a defined nucleosomal DNA state by preventing additional nucleosome assembly and remodeling.

Using purified ORC, Cdc6, Cdt1 and Mcm2-7 (*Kang et al., 2014*), we compared the ability of nucleosomal and naked DNA templates to participate in origin licensing as measured by loading of the Mcm2-7 helicase (*Figure 1B*). At the end of the reaction, DNA-beads were washed with a low-(L) or high-salt (H) containing buffer. The low-salt wash retains all DNA-associated proteins whereas the high-salt wash releases ORC, Cdc6, Cdt1 and incompletely-loaded Mcm2-7 but retains loaded Mcm2-7 complexes associated with successful origin licensing (*Donovan et al., 1997*; *Randell et al., 2006*). The amount of ORC DNA binding, helicase association (low-salt wash [L]) and helicase loading (high-salt wash [H]) were comparable between nucleosomal and naked DNA templates (*Figure 1C*). Thus, ISW1a-remodeled nucleosomes are permissive for origin licensing.

To address the effect of nucleosomes on origin selection, wild-type (WT) and mutant *ARS1*-containing DNA was assembled into nucleosomes and helicase loading was performed under lower-salt conditions that allow Mcm2-7 loading at non-origin sequences (compare upper and lower panels of (*Figure 1D*). Under these conditions, nucleosome assembly reduced non-specific Mcm2-7 loading onto mutant *ARS1*-containing DNA without altering helicase loading onto WT DNA (*Figure 1D*, top panel). Thus, nucleosomes reduced origin licensing at non-origin DNA sequences, consistent with

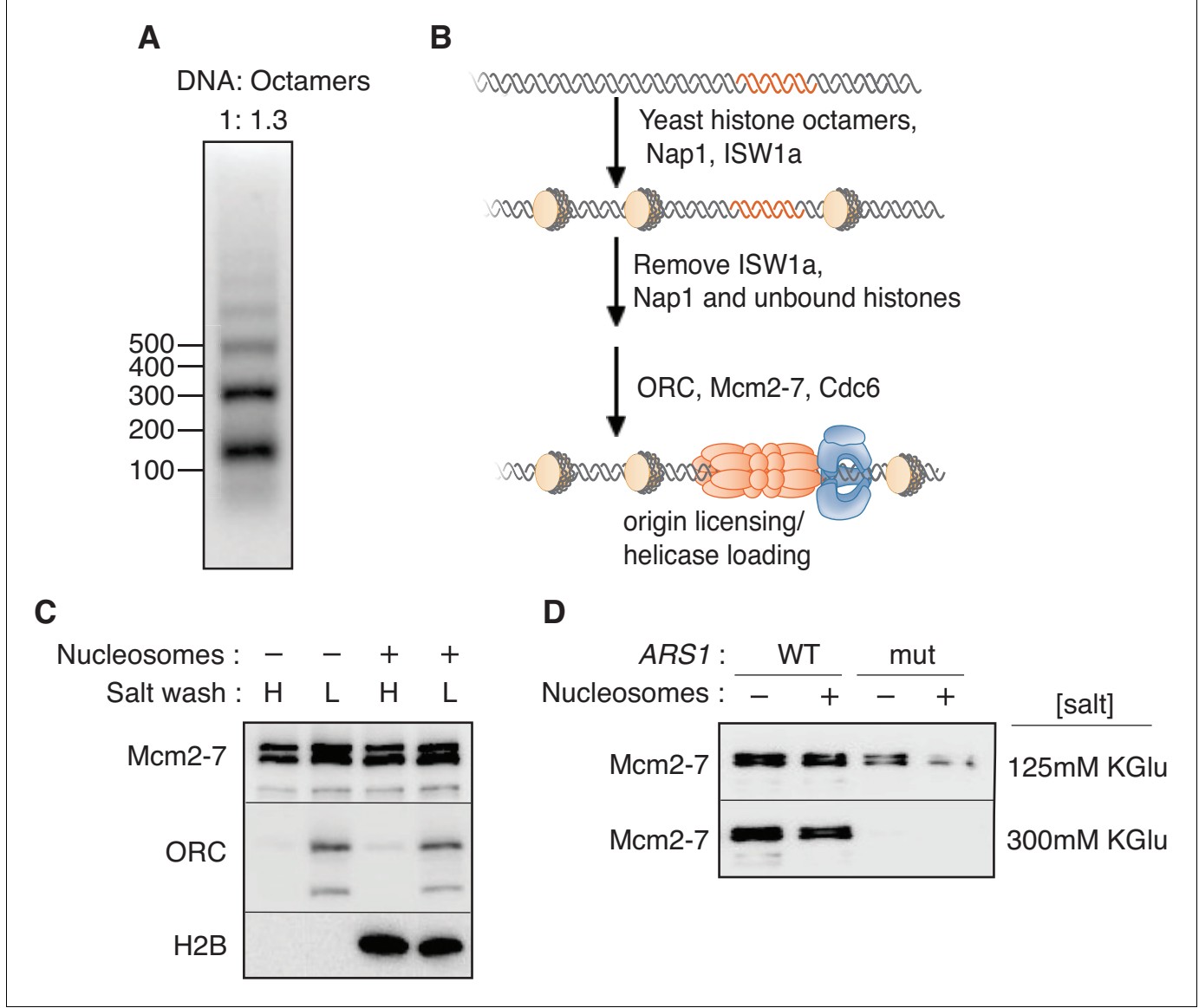

**Figure 1.** Mcm2-7 helicase loading onto nucleosomal DNA templates. (**A**) Nucleosomes were remodeled with bead-coupled ARS1-containing linear DNA, ISW1a, yeast histone octamers and Nap1. Nucleosome assembly was assessed after partial MNase digestion. (**B**) Outline of the helicase-loading assay using nucleosomal DNA. (**C**). Comparison of helicase loading on naked DNA and on ISW1a-remodeled nucleosomal DNA. DNA templates were washed with high-salt (**H**) or low-salt (**L**) buffer after loading. Template-associated Mcm2-7, ORC and H2B was detected by immunoblot. (**D**) Helicase loading onto either wild-type (WT) or A-B2- (mut) (*Heller et al., 2011*) ARS1-containing DNA. As indicated, nucleosomal DNA was remodeled with ISW1a. Assays were performed in either 125 mM (to allow increased origin non-specific helicase loading) or 300 mM (origin specific helicase loading) potassium glutamate. After a high salt wash, DNA-associated Mcm2-7 was detected by immunoblot.

The following figure supplements are available for figure 1:

**Figure supplement 1.** Purified proteins used in the in vitro nucleosome.

**Figure supplement 2.** Preparation of in vitro nucleosome templates.

**Figure supplement 3.** The ATPase activities of in vitro purified chromatin remodeling enzymes.

previous in vivo studies implicating local nucleosomes in origin selection (*Berbenetz et al., 2010*; *Eaton et al., 2010*).

## Origin-proximal nucleosome positioning influences helicase loading

To address how different local nucleosome landscapes influence replication initiation, we generated *ARS1* origin DNA templates with distinct nucleosome patterns. To this end, we assembled nucleosomes onto origin DNA in the presence of seven different purified CREs: ISW1a, ISW1b, ISW2, INO80-C, Chd1, SWI/SNF and RSC (*Figure 1—figure supplement 1B*). The amount of CRE added was normalized according to their relative ATPase activity (*Figure 1—figure supplement 3*), (*Smith and Peterson, 2005*). After nucleosome assembly, the CRE, Nap1 and non-nucleosomal histones were removed (*Figure 1—figure supplement 2B*) to ensure that the nucleosomes deposited during assembly are not remodeled or moved during subsequent replication-initiation assays. First, we examined nucleosome assembly by partial MNase-digestion. ISW1a, ISW1b, INO80-C, ISW2 and Chd1 each resulted in regularly-spaced nucleosomes on the origin DNA, albeit with different spacings (*Figure 2A*). In contrast, SWI/SNF- and RSC-remodeled nucleosomes did not show evidence of uniformly-spaced nucleosomes, consistent with previous observations (*Flaus and Owen-Hughes, 2003*; *Kassabov et al., 2003*). It was possible that SWI/SNF and RSC treatment reduced or eliminated nucleosome assembly. To test this hypothesis, we compared the amount of DNA-associated H2B and H3 (*Figure 2B* and *Figure 2—figure supplement 1A*) and the amount of mono-nucleosomal DNA produced after extensive MNase treatment (*Figure 2—figure supplement 1B*). These studies showed that the presence of different CREs did not dramatically change the extent of nucleosome formation. For simplicity, we refer to the different nucleosomal DNA templates by the CRE present during their assembly (e.g. SWI/SNF template).

We examined each of the different nucleosomal templates for origin licensing. SWI/SNF and RSC templates showed levels of Mcm2-7 loading similar to ISW1a templates (*Figure 2B*, *Figure 2—figure supplement 1C* and *Figure 2—figure supplement 1—source data 1* and *2*). ISW1b and INO80-C templates showed modest reductions in loaded Mcm2-7 and Chd1 and ISW2 templates showed progressively less loading. Thus, the CRE present during nucleosomal assembly impacted the extent of origin licensing.

## Local nucleosomes reduce helicase loading by inhibiting ORC DNA binding

To investigate the cause of the differential origin licensing, we determined the position of origin-proximal nucleosomes for the ISW1a, ISW1b, INO80-C, Chd1 and ISW2 templates using MNase-seq (*Cole et al., 2012*; *Eaton et al., 2010*). The ISW1a template showed a nucleosome-free region (NFR) overlapping *ARS1* with well-defined flanking nucleosomes (*Figure 2C*). In contrast, ISW2 template showed the appearance of a positioned nucleosome overlapping the origin (centered at $-54$ bp relative to ACS, *Figure 2C*). In addition, the flanking nucleosome on the opposite side of the origin was shifted towards the ACS (from +222 to +168) in the ISW2 templates. These data support a model in which encroachment of origin-proximal nucleosomes onto origin DNA directly inhibits origin licensing.

To determine whether the reduced origin licensing of the ISW2 and Chd1 templates was caused by decreased ORC DNA binding, we examined ORC association with these nucleosomal templates (*Figure 2D*). The extent of ORC binding to the ISW2 and Chd1 templates correlated with the amount of Mcm2-7 loading (*Figure 2B and D*). We asked if addition of ORC during nucleosome-assembly reactions restored Mcm2-7 loading. Importantly, when ORC bound DNA prior to Chd1- or ISW2-directed nucleosome assembly, loaded Mcm2-7 levels were restored to levels similar to ISW1a templates (*Figure 2E*). Together, these data indicate that nucleosome positioning over the origin reduces origin licensing by inhibiting ORC DNA binding and that ORC is not sufficient to move nucleosomes in the absence of a CRE.

To further investigate the role of ORC in the establishment of Mcm2-7-loading-competent chromatin states, we evaluated the role of the Orc1 bromo-adjacent homology (BAH) domain. BAH domains bind to nucleosomes (*Yang and Xu, 2013*) and elimination of the Orc1 BAH domain reduces initiation from a subset of replication origins in yeast (*Müller et al., 2010*). We purified ORC lacking the Orc1 BAH domain (ORCΔBAH, *Figure 2—figure supplement 2A*) and performed

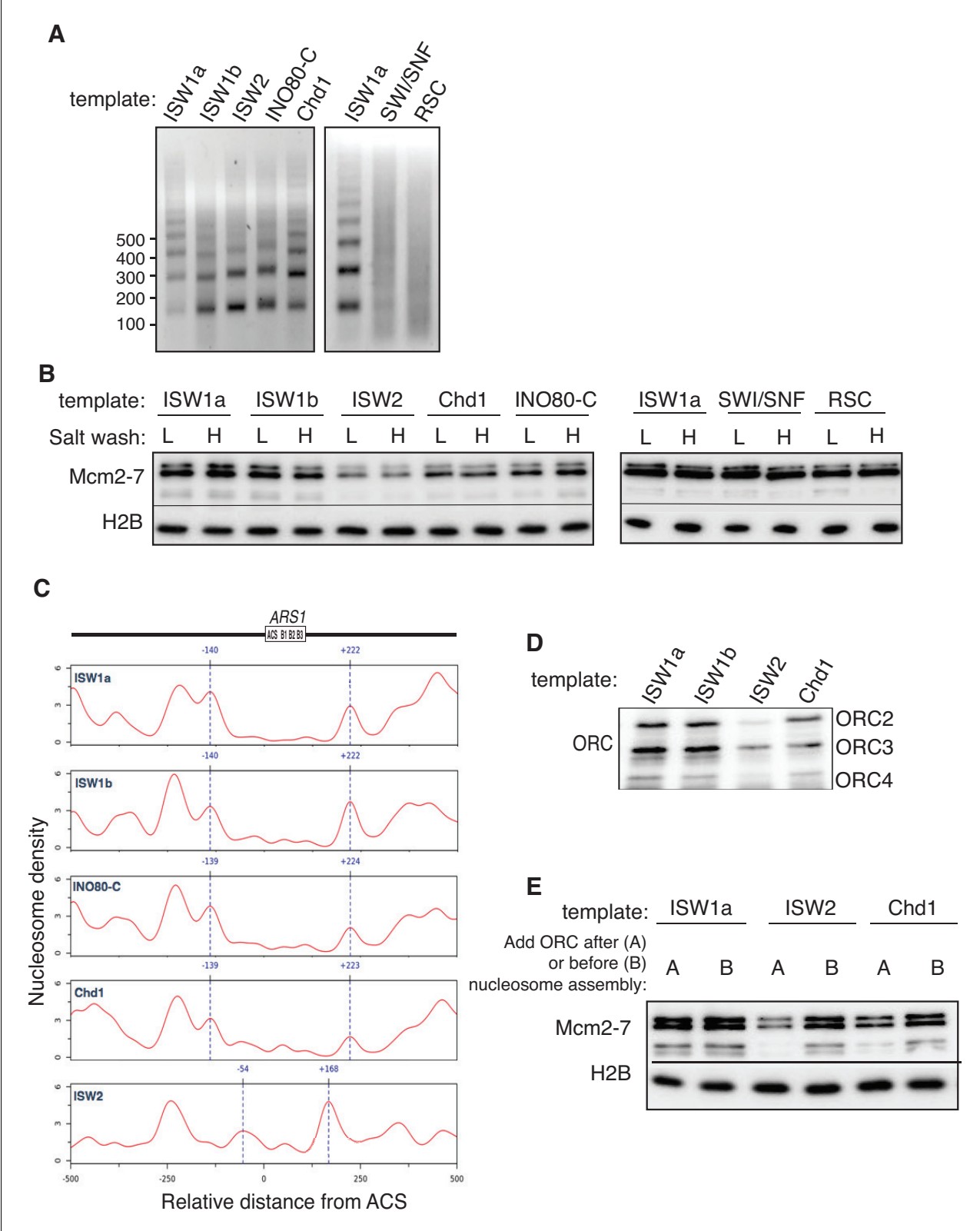

**Figure 2.** Comparison of helicase loading onto nucleosomal DNA templates remodeled with different CREs. (**A**) Comparison of nucleosome assembly with different CREs. Nucleosomes were remodeled with the indicated CRE and assayed by partial MNase digestion. (**B**) Helicase loading onto nucleosomes remodeled with different CREs. After helicase loading, DNA was washed either with high-salt (**H**) or low-salt (**L**) buffer. Mcm2-7 and H2B DNA association was detected by immunoblot. (**C**) Comparison of origin-proximal nucleosome positioning established by different CREs. The positions

*Figure 2 continued on next page*

*Figure 2 continued*

of nucleosome dyads remodeled with the indicated CRE were analyzed by high-throughput MNase-Seq. Nucleosome dyad density (Y-axis) and the corresponding position of the dyad (X-axis) are plotted. Zero on the X-axis indicates the first nucleotide of the *ARS1* consensus sequence (ACS). The elements of ARS1 (*Marahrens and Stillman, 1992*) are indicated above. (**D**) ORC association with nucleosomal DNA remodeled with different CREs. Template association of ORC was detected by immunoblot. (**E**) Addition of ORC during nucleosome assembly restores helicase loading on ISW2 and Chd1 templates. Nucleosomes were assembled onto *ARS1* DNA with the indicated CRE in the presence or absence of ORC. Helicase loading was performed and analyzed as described in (**B**).

The following source data and figure supplements are available for figure 2:

**Figure supplement 1.** Nucleosome assembly with different CREs and their ability to load Mcm2-7 helicase.

**Figure supplement 1—source data 1.** Raw values used in the quantification of *Figure 2B*, left panel (n = 3).

**Figure supplement 1—source data 2.** Raw values used in the quantification of *Figure 2B*, right panel (n = 3).

**Figure supplement 2.** ORC1 BAH domain and Abf1 is dispensable for helicase loading of nucleosomal templates.

helicase-loading assays using ISW1a, ISW2 and INO80-C templates (*Figure 2—figure supplement 2B–C*). Consistent with the limited effect of deletion of the Orc1 BAH domain on *ARS1* function in vivo (*Müller et al., 2010*), ORC and ORCΔBAH showed comparable levels of helicase loading onto all the nucleosomal templates.

We also examined whether the presence of the *ARS1*-binding protein, Abf1, influenced helicase loading in the presence of nucleosomes. Previous studies showed that Abf1 and ORC position nucleosomes on either side of *ARS1* (*Lipford and Bell, 2001*) and that elimination of the Abf1 binding sites reduced *ARS1* function (*Marahrens and Stillman, 1992*). Addition of purified Abf1 (*Figure 2—figure supplement 2D*) to either naked DNA or ISW1a templates did not improve helicase loading (*Figure 2—figure supplement 2E*). We also asked whether addition of Abf1 to the ISW2 nucleosome assembly would rescue the helicase-loading defects of ISW2 templates, as we observed for ORC (*Figure 2E*). In contrast to ORC, Abf1 did not improve helicase loading on the ISW2 template (*Figure 2—figure supplement 2F*).

## Local nucleosomes impact replication events downstream of helicase loading

Next, we examined the effect of nucleosomes on replication-initiation events after origin licensing had occurred. To this end, we performed replication assays (*Gros et al., 2014*; *Heller et al., 2011*; *On et al., 2014*) by sequentially adding DDK and an S-phase extract to helicases loaded onto DNA templates with or without nucleosomes (*Figure 3A*). ISW1a templates showed comparable levels of replication products to that of naked DNA (*Figure 3B*). Nucleotide incorporation was Cdc6- (*Figure 3B*), DDK- (*Figure 3C*, and *Figure 3—source data 1*) and origin-sequence-dependent (*Figure 3—figure supplement 1*) indicating that the DNA synthesis observed was due to replication initiation and elongation (rather than DNA repair).

To compare replication initiation from nucleosome templates remodeled with different CREs, we carried out the same replication assay with each template. For ISW1a, ISW1b, ISW2, INO80-C and Chd1, the level of replication products closely matched the amount of helicase loading with the same templates (compare *Figure 3C* and *Figure 2—figure supplement 1C*). Thus, origin activation and replisome assembly were not further reduced by these nucleosomal templates. In contrast, the RSC and SWI/SNF templates showed a disconnect between the extent of origin licensing and the levels of replication initiation. SWI/SNF, RSC and ISW1a templates showed comparable levels of Mcm2-7 loading (*Figure 2B* and *Figure 2—figure supplement 1C*), but the amount of replication products from SWI/SNF and RSC templates was reduced ~5 fold relative to ISW1a templates (*Figure 3D* and *Figure 3—source data 2*). Thus, nucleosomal DNA templates remodeled by SWI/SNF and RSC inhibit one or more events downstream of origin licensing.

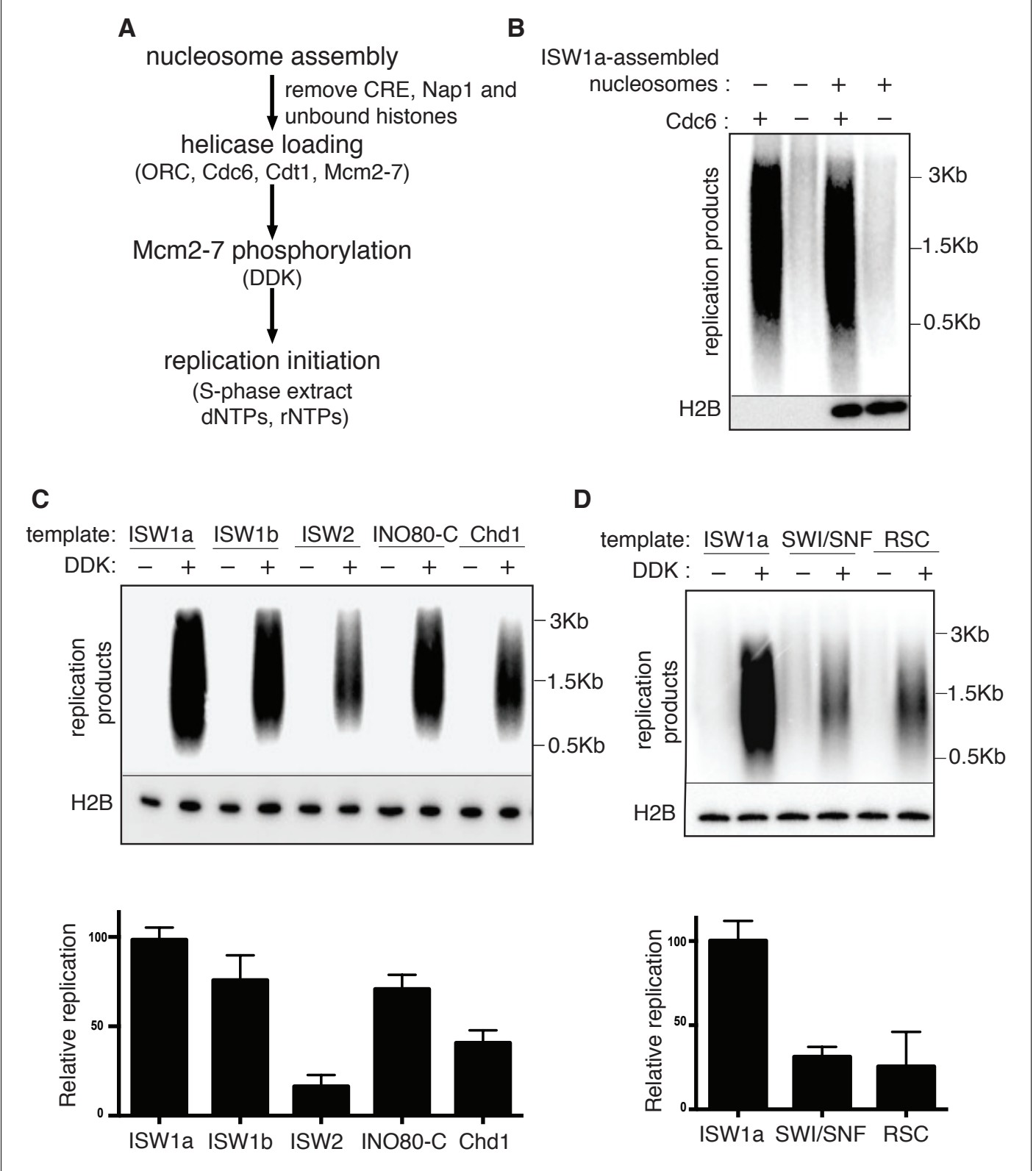

**Figure 3.** Replication initiation on nucleosome templates. (**A**) Outline of nucleosomal DNA replication initiation assay using purified proteins and yeast S-phase cell extract. (**B**) ISW1a templates do not interfere with replication. Naked DNA or ISW1a templates were assayed in the presence or absence of Cdc6. Radiolabeled replication products were analyzed by alkaline agarose electrophoresis and autoradiography (top). Template-associated H2B was detected by immunoblot (lower). (**C**) Comparison of replication using ISW1a, ISW1b, ISW2, INO80-C and Chd1 templates in the presence and absence

*Figure 3 continued on next page*

**eLIFE** Research article

Biochemistry | Genes and Chromosomes

*Figure 3 continued*

of DDK. Products of the extract-based replication assays were analyzed as in (B, top). H2B levels for each template are shown (middle). Quantification of replication products was performed as in *Figure 2B*. Error bars show the SD (n = 3, lower). (D) Comparison of replication of ISW1a, SWI/SNF and RSC templates in the presence or absence of DDK. Analysis of replication products, template-associated H2B and quantification (n = 3) as in (C).

The following source data and figure supplement are available for figure 3:

**Source data 1.** Raw values used in the quantification of *Figure 3C* (n = 3).
**Source data 2.** Raw values used in the quantification of *Figure 3D* (n = 3).
**Figure supplement 1.** In vitro nucleosomal DNA template replication initiation is origin specific.

## RSC and SWI/SNF templates impede CMG complex formation

The presence of multiple CREs in the S-phase extract led us to adapt a fully-reconstituted replication-initiation assay (*Lõoke et al., 2017*; *Yeeles et al., 2015*) to investigate the cause of the reduced replication of the SWI/SNF and RSC templates (*Figure 4A*). Compared to the S-phase-extract-based assay, ISW1a templates showed reduced replication using the fully-reconstituted assay (*Figure 4— figure supplement 1A*), most likely due to a lack of CREs and histone chaperones present in the S-phase-extract-based assay (*Devbhandari et al., 2017*; *Kurat et al., 2017*). Nevertheless, replication of the SWI/SNF and RSC templates was similarly reduced relative to their ISW1a-remodeled counterpart using the reconstituted assay (*Figure 4B*, *Figure 4—figure supplement 1B* and *Figure 4—figure supplement 1—source data 1*). Importantly, the reduced replication observed for the RSC or SWI/SNF templates was not simply because of a lack of uniformly-spaced nucleosomes. When we assembled nucleosomes in the absence of any CRE, the resulting nucleosomes were similarly non-uniformly spaced (*Figure 4—figure supplement 2A*) but the levels of replication from these templates were comparable to ISW1a templates (*Figure 4—figure supplement 2B*). Thus, the reduced replication capacity of the RSC and SWI/SNF templates requires the activity of the corresponding CRE.

To identify the replication event(s) that was reduced by SWI/SNF- and RSC-remodeled nucleosomes, we monitored different events of origin activation. First, we examined DDK phosphorylation of Mcm2-7 (detected by retardation of Mcm6 electrophoresis, *Francis et al., 2009*). This modification was either unchanged (SWI/SNF) or improved (RSC) relative to ISW1a templates (*Figure 4C*), indicating Mcm2-7 phosphorylation by DDK was not reduced. Next, we assessed CMG formation by examining Cdc45 and GINS template association after replication initiation and elongation (*Figure 4D*). Both SWI/SNF and RSC templates showed reduced Cdc45 and GINS template association compared to ISW1a templates. For these initial experiments, we measured template association at the end of the replication reaction. Thus, the decreases in Cdc45 and GINS template association could be due to inefficient CMG formation during initiation or increased CMG dissociation during elongation. To distinguish between these possibilities, we repeated the replication-initiation assays in the presence of ATP but without other rNTPs or dNTPs (*Figure 4E*). Under these conditions, the CMG can form and partially unwind DNA but replication cannot initiate (*Yeeles et al., 2015*). As in the previous assays, we observed reduced Cdc45 and GINS association with SWI/SNF and RSC templates compared to the ISW1a templates. Consistent with reduced active helicases and DNA unwinding, the amount of Rfa1 (a subunit of the eukaryotic single-stranded DNA binding protein RPA) association with RSC and SWI/SNF templates was also reduced (*Figure 4E*). Thus, the observed reduction in DNA replication products observed for the SWI/SNF and RSC templates in the complete assays was due to reduced CMG formation and helicase activation.

## CRE-specific restoration of replication and CMG formation to RSC and SWI/SNF templates

Our previous replication assays were performed in the absence of CREs to address how different chromatin states impact replication initiation. In vivo, however, these enzymes could be present at origin-proximal chromatin during initiation. To address whether the continuous presence of a CRE during replication initiation altered our findings, we asked if the addition of ISW1a, RSC or SWI/SNF

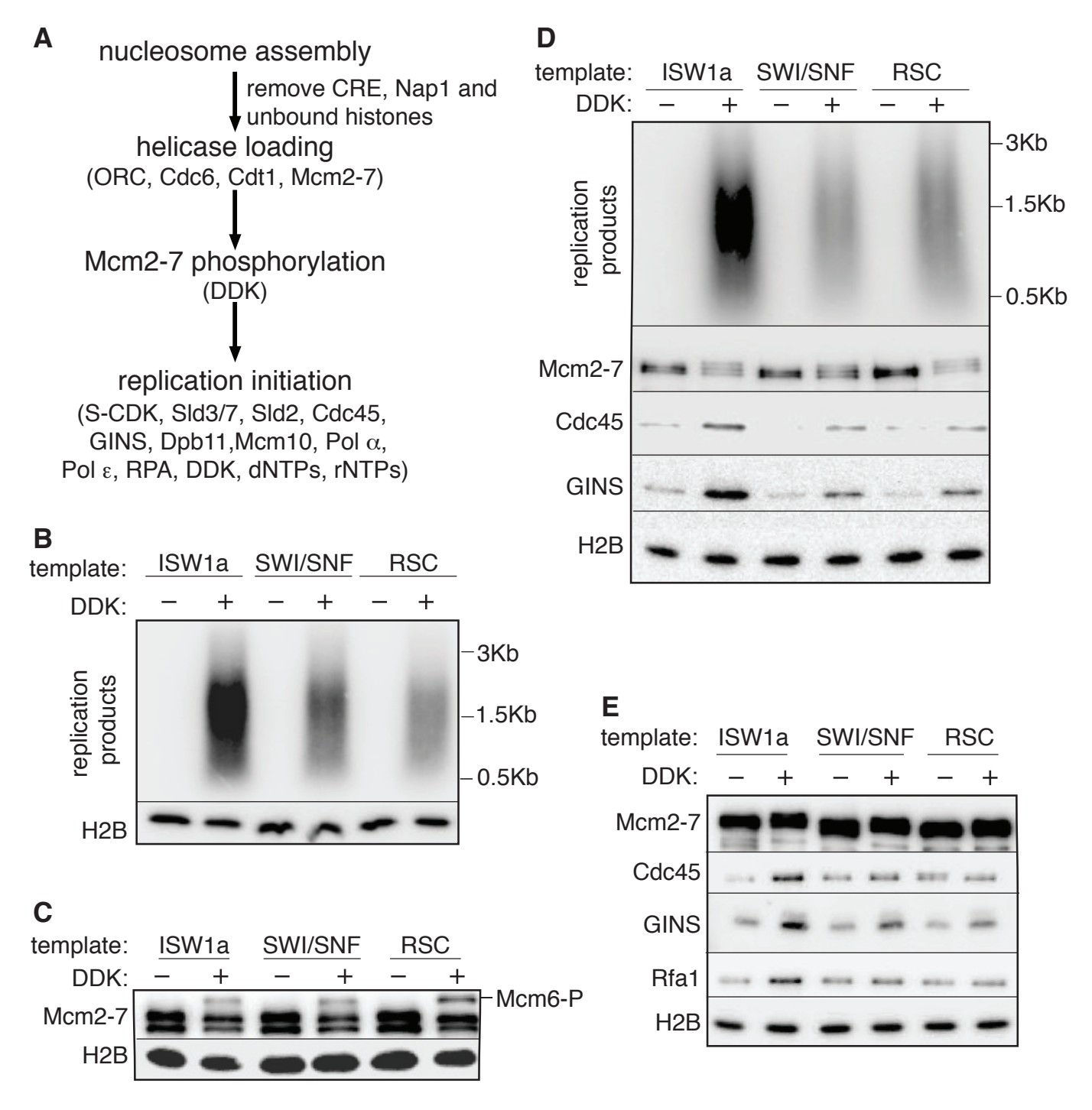

**Figure 4.** SWI/SNF and RSC templates show reduced CMG formation. (**A**) Outline of fully-reconstituted nucleosomal DNA replication initiation assay. The proteins added at each step are indicated. (**B**) Comparison of reconstituted nucleosomal DNA replication using ISW1a, SWI/SNF and RSC templates in the presence or absence of DDK. Analysis of replication products and H2B as in *Figure 3B*. (**C**) Comparison of Mcm2-7 phosphorylation by DDK on ISW1a, SWI/SNF and RSC templates. Phosphorylation of Mcm6 is indicated by reduced electrophoretic mobility and was analyzed by immunoblot (top). Template associated H2B is shown (lower). (**D**) Comparison of replication of ISW1a, RSC and SWI/SNF templates. Reactions were performed with or without DDK and replication products of the reconstituted replication reactions were analyzed as in *Figure 3B* (top). Template association of Mcm2-7, Cdc45, GINS and H2B was measured after a high-salt wash at the end of reconstituted replication assay by immunoblot (lower panels). (**E**) Comparison of CMG formation and activation using ISW1a, SWI/SNF and RSC templates. To prevent replication initiation, the only

*Figure 4 continued on next page*

*Figure 4 continued*

nucleotide present was ATP and Pol α was left out of the assay. Template association of Mcm2-7, Cdc45, GINS, Rfa1 and H2B were measured by immunoblot.

The following source data and figure supplements are available for figure 4:

**Figure supplement 1.** Reconstituted replication assay.

**Figure supplement 1—source data 1.** Raw values used in the quantification of *Figure 4B* (n = 3).

**Figure supplement 2.** Nucleosomal template assembled without CRE are able to replicate.

during replication-initiation assays improved replication initiation from the RSC and SWI/SNF templates. Adding ISW1a during the helicase-loading step (DL) of the assay (*Figure 5A*) restored CMG formation and increased DNA replication of the RSC templates (*Figure 5B*, compare lanes 4 and 5). In contrast, adding RSC during the helicase-loading step did not alter either the amount of replication or CMG formation (*Figure 5B*, lane 6). Similar experiments with SWI/SNF templates showed that the addition of ISW1a (but not SWI/SNF) improved replication of SWI/SNF templates (*Figure 5— figure supplement 1A*). In contrast to the ability of ISW1a to improve replication from the RSC and SWI/SNF templates, addition of RSC or SWI/SNF to ISW1a templates during helicase loading did not reduce replication levels (*Figure 5—figure supplement 1B*). Interestingly, consistent with its ability to reduce helicase loading, addition of Chd1 to ISW1a during helicase loading did reduce replication. Together these data indicate that the defects that we observe in CMG formation and replication for the RSC and SWI/SNF templates are not simply due to the lack of a CRE during the replication assay. Instead, our findings suggest that specific CREs create nucleosomal states that facilitate CMG formation and replication initiation.

Because we observed a connection between origin-proximal nucleosome positioning and origin-licensing capacity (*Figure 2C*), we asked if the reduced origin activation of SWI/SNF and RSC templates corresponded to a particular positioning of origin-proximal nucleosomes. Analysis of local nucleosome positioning by MNase-seq did not reveal a nucleosomal pattern that distinguished the RSC and SWI/SNF templates from ISW1a templates in origin proximal region (*Figure 5—figure supplement 2*). Consistent with the robust helicase loading observed for all three templates, they each exhibited a nucleosome-free region over the origin. The nucleosome pattern near *ARS1* was similar between the ISW1a and SWI/SNF templates. In addition, treatment of SWI/SNF templates with ISW1a did not cause major changes in nucleosome positioning. The pattern of RSC-remodeled nucleosomes was substantially different from SWI/SNF and ISW1a templates on the right side of *ARS1*. ISW1a addition to RSC templates enhanced the positioning of one nucleosome centered at ~400 bp on the right side of the *ARS1* ACS. Thus, unlike the situation for origin licensing (*Figure 2*), there was no apparent correlation between flanking nucleosome positions and the reduced origin activation observed for SWI/SNF and RSC templates.

Previous studies have reported a histone-binding motif in Mcm2 (*Foltman et al., 2013*; *Huang et al., 2015*), raising the possibility that nucleosome-Mcm2-7 interactions may facilitate replication initiation in a nucleosomal context. To test this possibility, we purified a mutant version of Mcm2-7 that lacks the Mcm2 histone-binding motif. Incorporation of this mutation did not alter helicase loading or DNA synthesis with or without nucleosomes (*Figure 5—figure supplement 3A–B*), suggesting that this interaction is not critical for replication initiation under the conditions of these assays. This is consistent with the lack of an obvious replication phenotype for this mutation (*Foltman et al., 2013*; *Huang et al., 2015*).

Given the redundant functions of chromatin remodelers in vivo, we asked if the ability to improve the replication of RSC and SWI/SNF templates was unique to ISW1a or if other CREs could perform the same function. As discussed above, addition of RSC or SWI/SNF to the corresponding nucleosomal templates during helicase loading did not improve CMG formation or replication (*Figure 5B* and *Figure 5—figure supplement 1*). Similarly, ISW2 addition to RSC templates resulted in only limited rescue of both CMG formation and replication (*Figure 5C*). In contrast, addition of ISW1b or INO80-C to RSC templates after helicase loading improved replication initiation and CMG formation

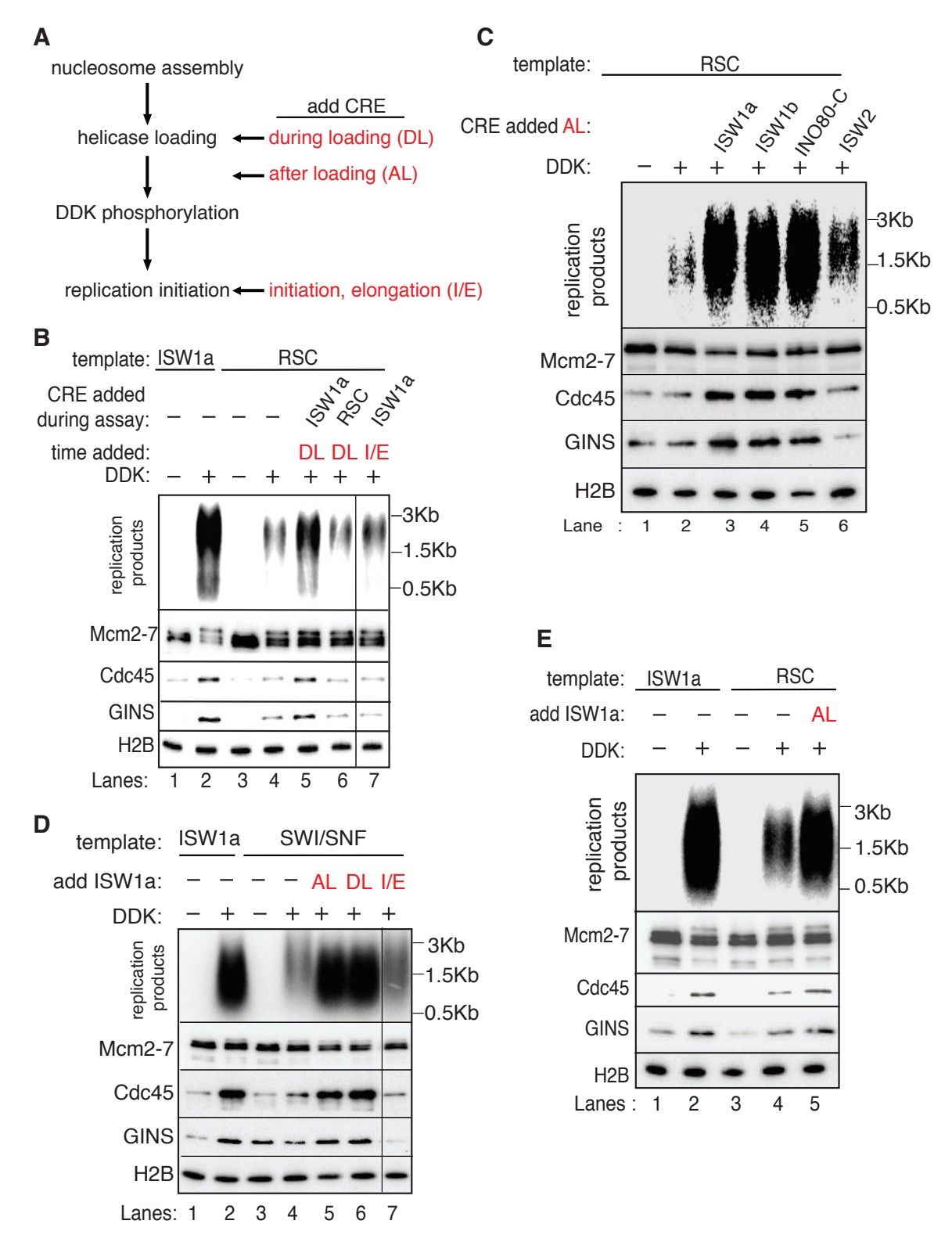

**Figure 5.** Rescue of SWI/SNF and RSC template replication initiation. (**A**) Schematic of ISW1a addition at various steps during the replication assay. (**B**) Addition of ISW1a at the helicase-loading step rescues replication initiation from RSC templates. Reconstituted replication assays were performed on ISW1a and RSC templates with or without DDK. ISW1a or RSC was added to the templates during helicase loading and not deliberately removed (DL) or upon addition of the helicase activation and elongation proteins (I/E) as indicated. The lane that show I/E is from the same gel as the rest of the

*Figure 5 continued on next page*

*Figure 5 continued*

panel. Replication products (top panel) and Mcm2-7, Cdc45, GINS and H2B template association (lower panels) were assayed as in *Figure 4D*. (C) Specific CREs improve RSC-template replication. Reconstituted replication assays were performed with RSC templates with or without DDK. ISW1a, ISW1b, INO80-C or ISW2 was added to RSC templates after helicase loading (AL). Replication products (top) and Mcm2-7, Cdc45, GINS and H2B template association (lower panels) were assayed as in *Figure 4D*. (D) Addition of ISW1a after nucleosome assembly facilitates replication and CMG formation of SWI/SNF templates. Reconstituted replication assays were performed with ISW1a or SWI/SNF templates with or without DDK. ISW1a was added to the templates either during helicase loading (DL), after helicase loading (AL) or upon addition of the helicase activation and elongation proteins (I/E) as indicated. The lane that show I/E is from the same gel as the rest of the panel. Replication products (top) and Mcm2-7, Cdc45, GINS and H2B template association (lower panels) were assayed as in *Figure 4D*. (E) ISW1a addition after helicase loading (AL) to RSC templates, but removed before helicase activation improves replication of and CMG complex formation on RSC templates. Reconstituted replication reactions were performed with the indicated templates with or without DDK. ISW1a was added to the RSC templates upon completion of helicase loading (AL). Replication products (top) and Mcm2-7, Cdc45, GINS and H2B template-association (lower panels) were assayed as in *Figure 4D*.

The following figure supplements are available for figure 5:

**Figure supplement 1.** ISW1a rescues RSC and SWI/SNF templates prior to the initiation step.

**Figure supplement 2.** Origin proximal nucleosome positioning is not directly responsible for CMG formation defects in RSC and SWI/SNF templates.

**Figure supplement 3.** Mcm2 histone-binding motif is dispensable for nucleosomal DNA replication.

**Figure supplement 4.** ISW1a rescues RSC and SWI/SNF templates after DDK step.

to similar levels as ISW1a addition (*Figure 5C*). Thus, the ability to restore full replication competence to the RSC or SWI/SNF templates is limited to a subset of CREs, consistent with previous studies indicating that these complexes have distinct functionalities (*Clapier and Cairns, 2009*).

We also asked when ISW1a needed to be present during a specific replication event to improve replication of SWI/SNF and RSC templates (*Figure 5A*). When added only during the helicase-loading step (DL) or after helicase-loading but removed before DDK treatment (AL), ISW1a significantly improved replication of the SWI/SNF (*Figure 5D*) and RSC templates (*Figure 5B and E*). Addition of ISW1a to the SWI/SNF and RSC templates after the DDK-step (but before CMG formation and replication initiation) also improved replication and CMG formation (*Figure 5—figure supplement 4*). In contrast, addition of ISW1a only during the initiation/elongation (I/E) step of the replication reaction did not improve CMG formation or replication of RSC (*Figure 5B*) or SWI/SNF templates (*Figure 5D*). Thus, ISW1a can only improve the replication competence of RSC and SWI/SNF templates if it acts prior to the events of origin activation.

## Discussion

The development of fully-reconstituted replication-initiation assays (*Yeeles et al., 2015*, *2017*) has opened the way to biochemically investigate the interactions between the replication-initiation machinery and nucleosomes. Although previous in vivo studies have revealed evidence that local chromatin states impact replication initiation (*Berbenetz et al., 2010*; *Eaton et al., 2010*; *Lipford and Bell, 2001*; *MacAlpine and Almouzni, 2013*; *Simpson, 1990*), a molecular understanding of these interactions has been difficult to attain. Here, we have used origin-containing nucleosomal DNA templates assembled in the presence of different CREs to investigate how origin-proximal nucleosomes affect replication initiation. For many of our studies, we have deliberately excluded CREs during the replication assays to investigate the impact of different static nucleosomal states on replication initiation. We also performed replication initiation assays in the presence of CREs, a situation that is potentially more representative of the in vivo situation at promoter-proximal origins. Together these studies reveal that local nucleosomes directly impact two steps in replication initiation: ORC DNA binding and CMG formation. Our studies do not include a full representation of the events that have the potential to impact replication initiation in vivo as they lack histone modifications and higher-order chromatin structures. Nevertheless, these studies represent an important first step to understanding the molecular mechanisms by which chromatin states modulate replication initiation.

## Specific CREs establish helicase-loading competent origin-proximal nucleosomes

Previous studies have shown that replication origins are included within nucleosome-free regions (NFRs) and this characteristic is important for origin activity (*Berbenetz et al., 2010*; *Eaton et al., 2010*; *Lipford and Bell, 2001*; *MacAlpine et al., 2010*; *Simpson, 1990*; *Xu et al., 2012*). Consistent with nucleosomes impacting origin selection, we found that assembly of DNA into nucleosomes reduced origin-independent initiation (*Figure 1D*). Given the redundancy of CREs in vivo, which CREs are capable of establishing NFRs at replication origins is unknown. Our findings demonstrate that CRE-dependent differences in local nucleosomes impact origin licensing. Only a subset of CREs positioned nucleosomes in a manner that allowed efficient origin licensing (*Figures 2B*, *6A and B*). ISW2 templates showed the most inefficient Mcm2-7 loading compared to other templates and this reduction correlated with the encroachment of origin-proximal nucleosomes over origin DNA in a manner that inhibited ORC DNA binding (*Figures 2C* and *6C*). This finding is consistent with studies showing that ISW2 slides nucleosomes towards the promoter-proximal NFR to suppress transcription at cryptic transcription-start sites (*Whitehouse et al., 2007*). Our findings suggest that ISW2 and perhaps Chd1 play a similar role in regulating origin usage. Interestingly, once ORC is bound to origin DNA, ISW2 is unable to displace ORC with a nucleosome (*Figure 2D*). Similarly, once ISW2

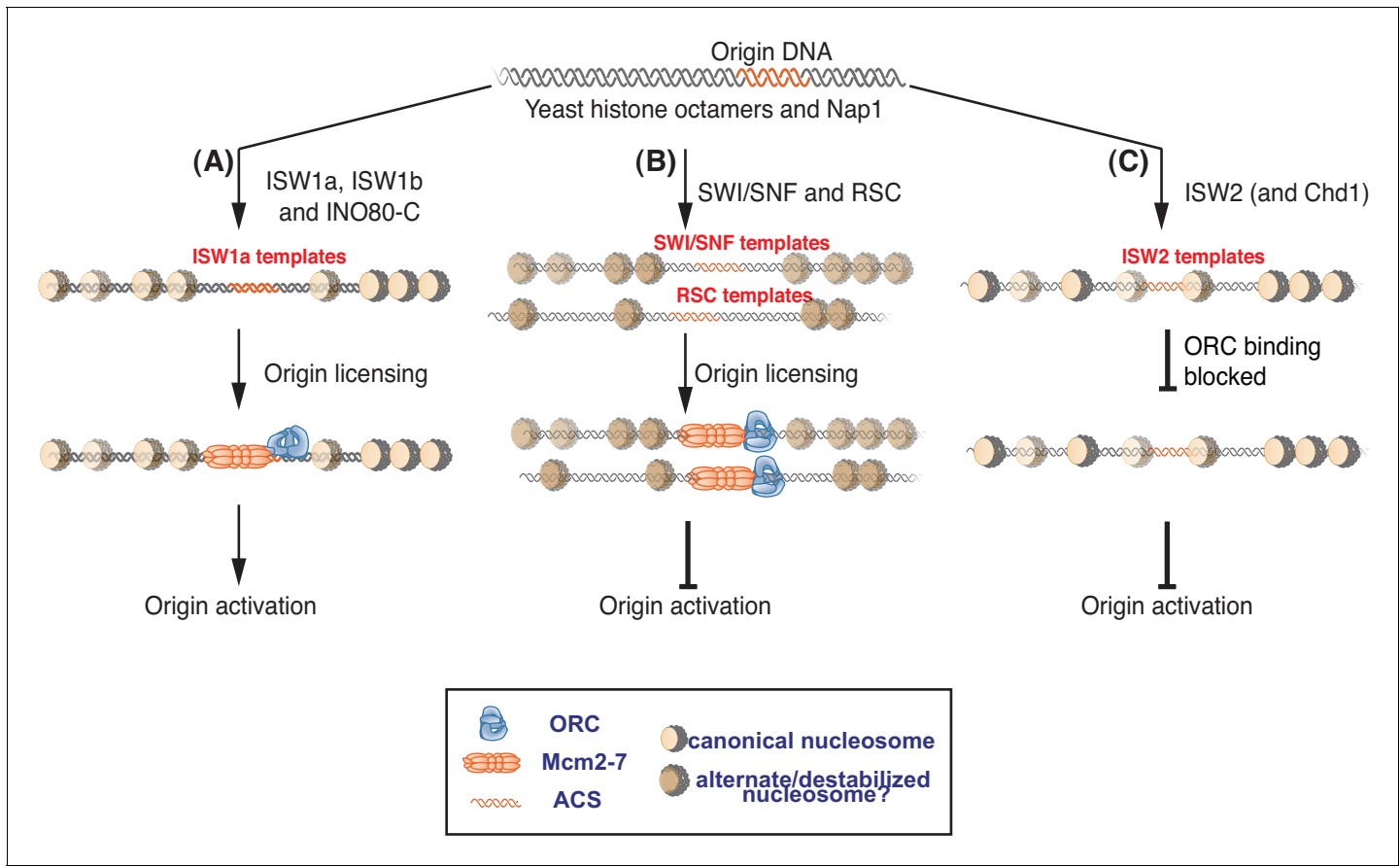

**Figure 6.** Nucleosomes remodeled by different CREs influence replication initiation differently. Nucleosomes affect multiple steps of replication initiation using distinct mechanisms. Schematic of ATP-dependent nucleosome assembly with different CREs and their affect on replication initiation. Opacity of the nucleosome represents nucleosome density at each location. (**A**) Replication permissive nucleosomes are remodeled by ISW1a, ISW1b and INO80-C. These templates are competent for both origin licensing and origin activation. Nucleosome positioning is comparable in these templates. (**B**) SWI/SNF and RSC templates are origin-licensing competent but are inefficient for subsequent origin activation. We propose that the SWI/SNF and RSC templates have alternate/destabilized nucleosome structures indicated by their different color and that these nucleosomes are not conducive to origin activation. Although both reduce origin activation, SWI/SNF and RSC templates do not share similar nucleosome positioning. (**C**) ISW2 (and Chd1) templates have nucleosomes over the replication origin that reduce ORC DNA binding and, therefore, origin licensing.

establishes nucleosome positioning at the origin, ORC is unable to bind (*Figure 2E*), suggesting ORC cannot displace interfering nucleosomes. These findings suggest that both the relative timing of ORC binding and histone deposition and the CRE present at this time will influence the use of a given site as an origin.

Neither the BAH domain of Orc1 nor the *ARS1*-binding protein Abf1 contributed to helicase loading in our experiments. The lack of a role for the ORC BAH domain is expected given the modest effect of deletion of the BAH domain on *ARS1* replication initiation in vivo (*Müller et al., 2010*). Given the observation that other BAH domains recognize specific modified forms of nucleosomes (*Yang and Xu, 2013*), it is also possible that we did not observe a role for the Orc1 BAH domain due to the unmodified status of the histones used in these experiments. Abf1 binding positions nucleosomes on one side of *ARS1* in vivo, however, ORC is able to perform this function in the absence of Abf1 at many origins (*Eaton et al., 2010*). One notable difference from the in vivo situation compared to our in vitro studies is that in vivo the *TRP1* gene transcribes into *ARS1*. Thus, it is possible that Abf1 binding is important to position nucleosomes in the presence of this invasive transcription but not in the absence (such as in our experiments).

## SWI/SNF and RSC templates reduce CMG formation

Our findings indicate that local nucleosomes also impact efficient CMG formation and, therefore, origin activation. In particular, SWI/SNF and RSC templates reduced this event. Our finding that the reduced replication of these templates is rescued by addition of other CREs (*Figure 5*) makes it clear that these effects are nucleosome-dependent. It remains to be determined how the RSC and SWI/SNF templates modulate this event. Given that these templates allow efficient helicase loading, simple steric inhibition by encroaching nucleosomes is unlikely to explain these effects. This conclusion is reinforced by the presence of a large nucleosome-free region overlapping the origin for both templates (*Figure 5—figure supplement 2*). Indeed, despite their different capacities for replication initiation and CMG formation, ISW1a and SWI/SNF templates had a similar pattern of surrounding nucleosomes and the SWI/SNF nucleosomes pattern was not changed by the same ISW1a addition that restored robust CMG formation and replication initiation to these templates (*Figure 5D* and *Figure 5—figure supplement 2*). Another possibility is that the lack of uniformly-spaced nucleosome arrays in the RSC and SWI/SNF templates reduces CMG formation. Of the seven CREs we tested, only SWI/SNF and RSC did not establish uniformly-spaced nucleosomes (*Figure 2A*). On the other hand, we observed a similar lack of uniformly positioned nucleosomes when we did not add any CRE to the nucleosome assembly reactions, and these templates showed much higher levels of replication initiation than the RSC and SWI/SNF templates (*Figure 4—figure supplement 2*).

Although it is possible that very subtle changes in nucleosome positioning are responsible, a more likely explanation is that the structure of the nucleosomes remodeled by RSC and SWI/SNF is different (*Figure 6A and B*). Previous studies have suggested that SWI/SNF family remodelers establish nucleosome structures that are different from canonical nucleosomes (*Lorch et al., 1998*; *Schnitzler et al., 1998*; *Ulyanova and Schnitzler, 2005*). Such altered nucleosomes could have distinct abilities to interact with the replication machinery. SWI/SNF and RSC are also known to remove H2A/H2B dimers from nucleosomes (*Clapier et al., 2016*). Although we do not observe a dramatic reduction in the relative amounts of H3 and H2A for these templates (*Figure 2—figure supplement 1A*) it is possible that a subset of nucleosomes assembled in the presence of RSC or SWI/SNF are lacking the full complement of H2A/H2B. One argument against this possibility is the ability of ISW1a addition to readily restore full replication initiation and CMG formation to these templates (*Figure 5*). Given that free histones are removed after initial nucleosome assembly, it is not clear how ISW1a addition could restore full nucleosomes to the RSC or SWI/SNF templates.

One interesting possibility to explain the different capacities of the templates to facilitate CMG formation is raised by recent studies suggesting that loaded Mcm2-7 interacts with adjacent nucleosomes (*Belsky et al., 2015*). It is possible that different positions/conformations of local nucleosomes (see below) impacts the ability of nucleosomes to interact with Mcm2-7 double hexamers. These interactions could directly or indirectly modulate access of helicase-activating proteins to loaded Mcm2-7 double hexamers. Addition of some CREs to RSC or SWI/SNF templates improved their replication but only if added before the helicase-activation proteins (*Figure 5*), suggesting that a positive interaction between loaded Mcm2-7 and nucleosomes must be established prior to CMG formation. Interestingly, we found that RSC and SWI/SNF could not reduce replication from an

ISW1a template when added during helicase loading. Perhaps a subset of CREs produce nucleosomes that can interact with the replication machinery positively and once these interactions are established they prevent other CREs from inducing alternative conformations. Although previous studies have identified a histone-binding motif in Mcm2 (*Foltman et al., 2013*; *Huang et al., 2015*), incorporation of this mutation into the Mcm2-7 complex did not alter helicase loading or DNA synthesis with or without nucleosomes (*Figure 5—figure supplement 3A–B*). It is possible, however, that this mutation is not sufficient to eliminate Mcm2-7 interactions with adjacent nucleosomes.

## CREs show different capacities to establish replication-competent chromatin states

Our studies show that the different CREs are not equivalent in their ability to establish replication-competent nucleosomes. These differences were observed both with regard to the initial deposition of nucleosomes on DNA (e.g. RSC templates inhibiting CMG formation, *Figure 5*) and when CREs were added after deposition (e.g. ISW1a addition rescuing the reduced CMG formation of RSC templates, *Figure 5*). The specificity of the different CREs in our assays suggests that the presence of different CREs at origins will impact origin usage. Localization of specific CREs to origin DNA through interactions with the replication machinery (*Euskirchen et al., 2011*; *Papamichos-Chronakis and Peterson, 2008*) or adjacent promoters/transcriptional machinery (*Yen et al., 2012*) could impact either origin licensing or activation. Our studies indicate that presence of a specific CRE at an origin during a particular cell cycle stage would have different consequences for DNA replication. A CRE present during S phase would impact the initial assembly of nucleosomes and more likely impact subsequent origin licensing/helicase loading. A CRE that is present during G1 is less likely to impact origin licensing and more likely to modulate subsequent CMG formation. Thus, simple deletion of a CRE or monitoring of CRE association with origin DNA in an asynchronous cell culture is unlikely to reveal their full impact on the events of DNA replication.

Interestingly, the RSC and SWI/SNF templates show hallmarks of late-initiating origins. Like late-initiating origins (*Belsky et al., 2015*; *Santocanale et al., 1999*; *Wyrick et al., 2001*), these templates showed efficient origin licensing but reduced/delayed replication initiation. In addition, replication timing is established in late M or early G1 phase (*Dimitrova and Gilbert, 1999*; *Raghuraman et al., 1997*) and replication timing can only be reprogrammed prior to S phase (*Peace et al., 2016*). Similarly, SWI/SNF and RSC templates can be remodeled to replicate efficiently if certain CREs are added prior to shifting the templates into helicase-activation conditions (*Figure 5B*), which is the biochemical equivalent of the G1-S phase transition. These similarities suggest that local nucleosome states influence replication timing.

Although our studies investigated DNA replication in the context of *S. cerevisiae* DNA-sequence-defined origins of replication, they are relevant to DNA replication in all eukaryotic organisms. Although most organisms do not use sequence-defined origins of replication, origin-proximal nucleosome-free regions are a common characteristic of origins in many organisms (*Fragkos et al., 2015*). Thus, our findings regarding the impact of local nucleosomes on origin licensing and selection are relevant to these origins as well. Indeed, the absence of specific sequences directing initiation of replication suggests that local chromatin states will have an even more important role in most organisms. Importantly, once helicases are loaded, specific origin sequences have little or no impact on subsequent origin activation (*Gros et al., 2014*, *2015*). The assays described here lay the groundwork for future studies of the impact of nucleosome structure and histone modification on DNA replication, and can be extended to query DNA replication-dependent nucleosome assembly events and epigenetic inheritance mechanisms.

## Materials and methods

### Construction of yeast strains and plasmids

Yeast strains used in these studies are derivatives of W303 and are described in *Supplementary file 1*. Epitope tagging was performed by PCR-based homologous recombination as previously described (*Longtine et al., 1998*). Plasmids used in this study are described in *Supplementary file 2* and were created by conventional molecular-cloning methods.

## DNA template preparation

DNA templates were isolated from the pARS1-Nco-Nco plasmid (*Kang et al., 2014*). The plasmid was digested with BamHI, filled in with biotinylated-dATP, dGTP, dCTP and dTTP using Klenow enzyme (NEB) to facilitate attachment to beads. After spin column purification (Plasmid Miniprep Kit from Qiagen), the biotin labeled DNA was cut with Nsi I and Sac II followed by a second spin column purification. This creates a large 3.8 kb BamHI to Nsi I DNA fragment that is biotinylated at one end and that is entirely derived from native *S. cerevisiae* sequences surrounding the *ARS1* origin of replication. A small biotinylated DNA (released by Sac II) is removed by the spin column and the remaining bacterial/vector DNA is not biotinylated and will not bind to streptavidin beads. The 3.8 kb biotinylated-DNA was immobilized on streptavidin-coated paramagnetic beads (Dynabeads M-280, ThermoFisher) according to manufacturer instructions and the non-biotinylated DNA fragment was washed away.

## Protein purification

Yeast histones and hNap1 were purified using previously established methods (*Vary et al., 2004*). Mcm2-7/Cdt1, ORC, Cdc6 and DDK were purified as previously described (*Heller et al., 2011*; *Kang et al., 2014*). S-CDK, Sld3/Sld7, Sld2, Cdc45, Dpb11, GINS, Pol ε, Pol α/primase, RPA and Mcm10 were purified as described previously (*Lõoke et al., 2017*).

### Chromatin remodeling enzymes (CREs)

Tandem affinity purification of ISW1a (TAP-Ioc3), ISW1b (TAP-Ioc2), Chd41 (TAP-Chd1), RSC (TAP-Rsc2), INO80-C (TAP-Ino80), and SWI/SNF (TAP-Swi2) was performed as described (*Watanabe et al., 2015*). ISW2 (FLAG-Isw2) was purified according to manufacturer's protocol (Sigma), except that E-buffer (20 mM HEPES [pH7.5], 350 mM NaCl, 10% glycerol, 0.1% Tween) was used during the entire purification. Purified proteins were concentrated with VIVASPIN concentrators (Sartorius, Sartorius, Germany) and dialyzed against E-Buffer with 1 mM DTT. The ATPase activity of each remodeling complex was determined as described (*Smith and Peterson, 2005*), and the concentration of each remodeling complex was estimated relative to a SWI/SNF standard.

### ORCΔBAH

ORCΔBAH was purified as previously described (*Frigola et al., 2013*) with the following changes. After the Q-sepharose column, pooled fractions of ORCΔBAH were applied to a Superdex 200 column equilibrated with buffer H (50 mM HEPES-KOH [pH7.6], 1 mM EDTA, 1 mM EGTA, 5 mM MgOAc, 10% glycerol) with 0.3 M potassium glutamate.

### Abf1

Abf1 was purified as previously described (*Eaton et al., 2010*).

## Nucleosome assembly

Nucleosomes were assembled as previously described (*Mizuguchi et al., 2012*). Nucleosome formation was optimized using a 3.8 kb *S. cerevisiae* DNA fragment (see below), *S. cerevisiae* histone octamers (using an DNA:octamer ratio of 1:1.3 by mass) and varying human Nap1 (hNap1) concentration. After determining an optimal histone octamer:hNap1 ratio, nucleosome assembly was further optimized by varying ISW1a concentration. Finally, after optimizing the Nap1 and ISW1a concentrations, the nucleosome assembly reaction was optimized for the DNA:octamer ratio. Consequently, nucleosomes were assembled with ~137 nM yeast histone octamers, 267 nM hNap1, 10 nM CRE (ISW1a or ISW1b or ISW2 or INO80-C or Chd1 or SWI/SNF or RSC) and 120 fmol Dyna bead-immobilized 3.8 Kbp *ARS1* DNA in a 20 μl reaction. Initially, hNap1, histone octamers were incubated in ExB 5/50 buffer (10 mM HEPES-KOH [pH7.6], 0.5 mM EGTA, 5 mM magnesium chloride (MgCl$_2$), 50 mM potassium chloride (KCl), 10% glycerol, 0.1 mg/ml BSA) in for for 45 min followed by CRE addition and continued incubation for another 15 min., bringing the total reaction volume to 12.5 μl. 7.5 μl of ATP regeneration mix (5 mM ATP, 30 mM creatine phosphate and 1 μg/ml creatine kinase in 1x ExB 5/50 buffer) was added to the histone octamers, hNap1 and CRE reaction mix and immediately added to the bead-immobilized DNA and incubated at 30℃ at 1400 rpm for 4.5 hr in a Thermomixer (Eppendorf, Hauppauge, NY). Nucleosomal-DNA beads were stored at 4℃ and used

for assays within 12–24 hr. Nucleosome assembly was analyzed by digesting 120 fmol nucleosomal DNA with limiting (0.04 U) MNase at 25°C at 1300 rpm for 15 min in a Thermomixer. The resulting DNA fragments were purified using spin columns (EZ Nucleosomal DNA prep Kit from Zymo Research) and separated on a 1.5% agarose gel and stained with ethidium bromide. Purification protocols for chromatin-remodeling enzymes, histones and hNap1 are described in *Supplementary file 2*.

## Helicase-loading assay

Helicase loading was performed as previously described (*Kang et al., 2014*) for naked DNA templates with the following modifications. Mcm2-7/Cdt1, ORC, and Cdc6 were purified as previously described (*Kang et al., 2014*). The bead-coupled nucleosomal DNA was magnetically separated from unassociated or loosely bound proteins and the supernatant was removed. Nucleosomal DNA was washed twice with 20 µl buffer A-0.35 (25 mM HEPES-KOH [pH7.6], 0.5 mM EGTA, 0.1 mM EDTA, 5 mM $MgCl_2$, 10% glycerol, 0.02% NP40, 0.1 mg/ml BSA and 0.35 M KCl) and once with 20 µl buffer A-0.3 KGlut (0.3 M potassium glutamate [KGlut] instead of 0.35 M KCl in buffer A-0.35). Helicase loading was initiated by the addition of 120 fmol ORC, 180 fmol Cdc6, and 360 fmol Mcm2–7/Cdt1 in a 20 µl reaction containing 60 fmol of bead-coupled 3.8 kb *ARS1* DNA (with or without nucleosomes) in helicase-loading buffer (25 mM HEPES-KOH [pH 7.6], 12.5 mM magnesium acetate (MgAc), 300 mM KGlut, 20 µM creatine phosphate, 0.02% NP40, 10% glycerol, 3 mM ATP, 1 mM dithiothreitol (DTT), and 2 µg creatine kinase). The reaction were briefly vortexed or mixed by pipetting (if necessary) to remove any bead clumping. The reactions were incubated at 25°C at 1250 rpm for 25 min in a Thermomixer. Beads were washed three times with 150 µl Buffer H (25 mM HEPES-KOH [pH 7.6], 1 mM EDTA, 1 mM EGTA, 5 mM MgAc, 10% glycerol, and 0.02% NP40) containing 0.3 M KGlut for low salt wash experiments. Experiments with high-salt washes substituted buffer H with 0.5 M NaCl for the second of the three wash steps. DNA-bound proteins were eluted from the beads using 2x sample buffer (120 mM Tris [pH 6.8], 4% SDS and 20% glycerol). Eluted proteins were separated by SDS-PAGE and analyzed by immunoblotting.

## Quantification of replication products

Replication products were measured by incorporation of $^{32}$P-dCTP into newly synthesized DNA. Incorporated $^{32}$P-dCTP was detected after denaturing gel electrophoresis using a phosphor-imager. For relative replication product quantification, nucleosomal templates assembled with the indicated CRE were quantified with ImageJ software. For each assay, three (n = 3) biological replicates were quantified. The mean value for ISW1a (reactions with DDK) was calculated and set as 100%. All the other values were calculated as a percentage of the mean value of the ISW1a experiment (always performed as part of the same experiment and separated on the same gel). Statistical analysis was performed using Prism software. Error bars indicate standard deviation (SD).

## Replication initiation assays

### Extract-based replication assay

Replication assays with extracts were performed as previously described (*Kang et al., 2014*). Nucleosomes were remodeled, washed and Mcm2-7 was loaded on 60 fmol of DNA in 20 µl reaction volume as described above. After helicase loading, the reaction mix was removed and loaded Mcm2-7 complexes were phosphorylated with 930 fmols DDK in DDK reaction buffer (50 mM HEPES-KOH [pH7.6], 3.5 mM MgAc, 225 mM KGlut, 0.02% NP40, 10% glycerol, 1 mM spermine, 1 mM ATP, and 1 mM DTT) in 30 µl. After DDK phosphorylation was completed, the reaction mix was removed from the beads and replication was initiated by adding 375 ng of S-phase yeast extract to replication buffer (25 mM HEPES-KOH [pH 7.6], 12.5 mM MgAc, 300 mM KGlut, 20 µM creatine phosphate, 0.02% NP40, 10% Glycerol, 3 mM ATP, 40 µM dNTPs, 200 µM CTP/UTP/GTP, 1 mM DTT, 10 µCi [α-$P^{32}$] dCTP, and 2 µg creatine kinase) in a final volume of 40 µl and incubated for 1 hr at 25°C and 1250 rpm in a Thermomixer. Upon completion, the nucleosomal DNA beads were washed as described for the helicase-loading assay using a low-salt wash. Occasional clumping of nucleosomal DNA beads was eliminated by vortexing. DNA synthesis was monitored using 0.8% alkaline-agarose (in 30 mM sodium hydroxide) gel electrophoresis followed by detection of incorporated $^{32}$P-dCTP.

DNA-bound proteins were released from the beads with 2X sample and analyzed by SDS-PAGE and immunoblotting.

## Reconstituted replication assay for nucleosomal DNA

The reconstituted nucleosomal DNA replication assay was adapted from a reconstituted replication assay described for naked DNA (*Lõoke et al., 2017*). Helicase-loading reactions were performed with 60 fmol of nucleosomal DNA (washed with buffer A-0.35 and buffer A-0.3 KGlut as previously described in the text) with 500 fmol ORC, 500 fmol Cdc6 and 500 fmol Mcm2-7/Cdct1 in 10 µl helicase-loading buffer at 25°C at 1250 rpm for 25 min in a Thermomixer. Upon completion of helicase loading, the DNA beads were isolated and the supernatant was removed. Next, 900 fmol of DDK in 10 µl DDK reaction mix (described above) was added to the DNA beads to phosphorylate the loaded Mcm2-7 complexes for 20 min at 25°C at 1250 rpm in a Thermomixer. Upon completion of DDK phosphorylation, the DDK reaction mix was removed from the beads and the replication initiation/elongation step was carried out by adding the indicated amounts of the following proteins (0.5 pmol S-CDK, 0.1 pmol DDK, 0.5 pmol Sld3/7, 2.5 pmol Cdc45, 1 pmol Sld2, 1 pmol Dpb11, 2.5 pmol GINS, 80 fmol Mcm10, 0.93 pmol Polε, 1.25 pmol Polα and 1 pmol RPA) in 30 µl of replication-initiation buffer (25 mM HEPES-KOH [pH7.6], 12.5 mM MgAc, 300 mM KGlut, 20 µM creatine phosphate, 0.02% NP40, 0.04 mg/ml BSA, 10% Glycerol, 3 mM ATP, 40 µM dNTPs, 200 µM CTP/UTP/GTP, 1 mM DTT, 10 µCi [α-P$^{32}$] dCTP, 2 µg creatine kinase, and 0.5X complete protease inhibitor [Roche]) to the DNA beads for 60 min at 25°C and 1250 rpm in a Thermomixer. At the end of the reaction, immobilized-DNA was washed with a high-salt wash as previously described. Replicated DNA and associated proteins were analyzed as previously described for extract-based replication assay. For assays which examined replication protein association prior to elongation, reactions were performed by leaving out dNTPs, rNTPs (except ATP) and Polα. For experiments in which a CRE was added during a specific step of the reconstituted replication assay, the indicated CRE was added to a final concentration of 10 nM. CRE addition was added at five different times during replication assay: (DL) the CRE was added during the helicase-loading step but was washed off at the end of this step; (AL) the CRE was added after helicase loading but washed away before DDK-phosphorylation; (After DDK) the CRE was added after DDK-phosphorylation but washed away before addition of helicase-activation and elongation proteins; (I/E) the CRE was added with elongation and helicase-activation proteins. When indicated, the CRE was removed by sequential 20 µl washes with A-0.35 KCl (1X) and A-0.3 KGlut (1X). Purification protocols for replication proteins are described in *Supplementary file 2*.

## Quantification of helicase-loading

Quantification of relative Mcm2-7 loading (immunoblots for Mcm2-7) for nucleosomal templates assembled with the indicated CRE was determined with ImageJ software. For each assay three (n = 3) biological replicates were quantified. The mean value for ISW1a (reaction with high-salt wash) was calculated and set as 100%. All the other values were calculated as a percentage of the mean value of the ISW1a experiment (always performed as part of the same experiment and separated on the same gel). Statistical analysis was performed using Prism software. Error bars indicate standard deviation (SD).

## Mono-nucleosome analysis

Nucleosomes were remodeled on 120 fmol of DNA as previously described (nucleosome assembly and analysis section) with ISW1a, SWI/SNF or RSC. Nucleosomal DNA was digested with 1 U of MNase for 30 min shaking at 25°C at 1350 rpm in Thermomixer. The released DNA was purified using spin columns (EZ Nucleosomal DNA prep Kit from Zymo Research). The purified DNA was separated on a 1.5% agarose gel electrophoresis and stained with ethidium bromide.

## Nucleosome positioning analysis

Nucleosomal DNA was washed with buffer A-0.35 (2X) and digested with 1 U of MNase in 20 µl MNase-digestion buffer (12.5 mM HEPES-KOH [pH7.6], 0.5 mM EDTA, 5 mM MgCl2, 2 mM calcium chloride, 5% glycerol and 1 mM DTT) for 30 min at 25°C shaking at 1350 rpm in a Thermomixer. DNA was purified using spin columns. The purified DNA was separated on a 1.5% agarose gel

electrophoresis and stained with ethidium bromide. DNA fragments in the size range that includes mono-nucleosomes (<160 bp) were extracted from the gel and purified using spin columns (Freeze 'N Squeeze DNA Gel Extraction Spin Columns, Bio-Rad). DNA samples were end-repaired and adaptor-ligated using the SPRI-works Fragment Library System I (Beckman Coulter Genomics) and indexed during amplification. Libraries were quantified using the Fragment Analyzer (Advanced Analytical) and qPCR before being loaded for paired-end sequencing using an Illumina HiSeq 2000 (MIT BioMicroCenter).

Paired-end sequencing reads were aligned to the plasmid sequence using Bowtie 1.1.2 (*Langmead et al., 2009*) with the following parameters: -n 2 -l 20 - -best - -strata. To obtain nucleosome occupancy profiles, the midpoint positions from all 125–175 bp read fragments were extracted. The nucleosome signal was smoothed by constructing a 20 bp Gaussian kernel around each midpoint position, and smoothed kernels were aggregated together to form a nucleosome signal track (*Boyle et al., 2008*). Signal was then normalized to read depth for each sample. Consensus nucleosome positions were determined by finding peaks (above a threshold of 2) in the nucleosome-density signal (*Flores and Orozco, 2011*).

# Acknowledgements

We thank Manabu Gaku Mizuguchi and Carl Wu for advice on nucleosome assembly assays and members of the Bell laboratory for helpful discussions. We thank Megan Warner for comments on the manuscript. This work was supported the by NIH grants GM054096 (CLP) and R01-GM104097 (DMM). SPB is an investigator with the Howard Hughes Medical Institute. IFA was supported in part by American Cancer Society Postdoctoral Fellowship (123700-PF-13-071-01-DMC). This work was supported in part by the Koch Institute Support Grant P30-CA14051 from the NCI. We thank the Koch Institute Swanson Biotechnology Center for technical support, specifically the Biopolymers and Genomics cores.

# Additional information

## Funding

| Funder | Grant reference number | Author |
| --- | --- | --- |
| American Cancer Society | 123700-PF-13-071-01-DM | Ishara F Azmi |
| National Institute of General Medical Sciences | GM54096 | David M MacAlpine Craig L Peterson |
| National Institute of General Medical Sciences | GM104097 | David M MacAlpine Craig L Peterson |
| Howard Hughes Medical Institute | Investigator | Stephen P Bell |

The funders had no role in study design, data collection and interpretation, or the decision to submit the work for publication.

## Author contributions

IFA, Conceptualization, Resources, Formal analysis, Methodology, Writing—original draft, Writing—review and editing; SW, MFM, SK, Resources, Methodology; JAB, Formal analysis, Writing—review and editing; DMM, CLP, Supervision, Writing—review and editing; SPB, Conceptualization, Funding acquisition, Writing—review and editing

## Author ORCIDs

Ishara F Azmi, http://orcid.org/0000-0001-6280-3149

Jason A Belsky, http://orcid.org/0000-0003-2945-6282

Stephen P Bell, http://orcid.org/0000-0002-2876-610X

## Additional files

**Supplementary files**

• Supplementary file 1. Yeast strains used in the study.

• Supplementary file 2. Plasmids used in the study.

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
