## [Decision Letter]

Thank you for submitting your article "Nucleosomes Influence Multiple Steps During Replication Initiation" for consideration by *eLife*. Your article has been reviewed by three peer reviewers, and the evaluation has been overseen by a Reviewing Editor and Jessica Tyler as the Senior Editor. The reviewers have opted to remain anonymous.

The reviewers have discussed the reviews with one another and the Reviewing Editor has drafted this decision to help you prepare a revised submission.

Summary:

The proposed studies are clearly very early work on chromatin and initiation of DNA replication in vitro. The studies represent a very good first step in understanding a complex mechanism, even if the detailed molecular insight remains to be determined. It will be acceptable for publication provided they address the following important criticisms and adequately reply to these comments:

Essential revisions:

1) They demonstrate the activity of each CRE in an ATPase assay or on a mononucleosomal template.

2) They test whether the presence of a histone chaperone has an effect on the replication assays, where histones are presumably displaced.

3) They provide a better description of how the chromatin was prepared.

4) They discuss the caveat of why the specificity of initiation seen in vivo has not been duplicated using their in vitro system.

Other Comments:

1) The manuscript does not mention one of the main interaction points between the chromatin and the initiation machinery: the BAH domain of ORC1 that can bind nucleosomes. This domain has been shown in yeast to affect certain initiation events and in humans this domain has been implicated in both reading an origin "histone code" and potentially in Meier-Gorlin Syndrome. A mutant version of ORC could be produced with the BAH removed and the assays in this manuscript could be repeated. If the domain is not important for any activities described in the paper, this may suggest that post-translational modifications of the histones may be required for BAH-specific activity. Alternatively, the BAH domain may be necessary for ORC, when pre-bound to naked DNA, to resist the conversion of the chromatin organization by ISW2 to a non-permissive state.

2) SWI/SNF and RSC CREs generate chromatin organization that is not permissive to origin activation, yet it is not clear why this occurs. It is also not clear why subsequent treatment with Isw1a rescues activation. The nucleosome occupancy patterns in Figure 5—figure supplement 2 do not shed light on this. One possibility is that there are altered nucleosomes around the origin that are formed by SWI/SNF or RSC that are not adequately profiled by the MNase-Seq, which is described as sequencing only mono-nucleosomal fragments less than 160 bp. Alternatively, is there a possibility that the intensity of the nucleosomes is not easily comparable across different templates, due to internal normalization? The profiles of the whole 3.8 kb template for each remodeling condition may help to address this, or using some independent method to fragment the DNA and sequencing could address percent occupancy. Addressing these points would at the very least rule out certain caveats.

3) Why do you need to purify away the Nap1 and CREs? This is not discussed, but I would argue that you may need a histone binding protein to avoid that the evicted histones precipitate onto DNA again. How much of each CRE remains bound is a critical issue.

4) In Figure 1—figure supplement 3, are anti-CPB and anti-FLAG on the same membrane? If not, an internal reference should be used to validate the wash of the CREs. The homogenous removal of the CREs is essential in interpreting the results.

5) Clarify these statements:- "The results of the MNAse in presence of SWI/SNF and RSC (Figure 2) shows that the "nucleosomes" in those samples are not stable". Could you perform the MNase-seq (Figure 5—figure supplement 2) and get a 147 bp protection in Figure 2—figure supplement 2? What is the difference between these experiments?

6) Figure 1 should include a Western blot for ORC as well.

7) Figure 2 would be improved by showing ORC occupancy in all CRE contexts.

8) The size range of all replication products run on denaturing agarose gels should be indicated with marker indicator.

9) Dissection of the steps after DDK treatment, using purified components such as Cdc45, would make this manuscript even more impactful.

10) Including a mutant MCM2 lacking the histone binding domain would increase the impact of the manuscript, similar to the ORC BAH mutant.

11) Including the Abf1 protein, which binds the B3 element of ARS1, would increase the impact of the manuscript. Previously it was proposed that Abf1 helps establish a nucleosome free region for MCM loading.

We have included the full reviews at the bottom of this letter, to further clarify these requests.

Reviewer #1:

This manuscript addresses the effects of chromatin on the complete mechanism of the initiation of DNA replication in a reconstituted system for the first time, using the well-established budding yeast model. The authors first show that a regularly-spaced nucleosome array made with recombinant histones is equally efficient at one of the first major steps in initiation, the formation of the pre-replication complex (pre-RC), which culminates in the loading of the MCM helicase rings. Next, they prepare nucleosomal templates using multiple chromatin remodeling enzymes that generate different patterns of nucleosome placement. These different nucleosome organizations result in different efficiencies of pre-RC formation and further analysis suggests that the positioning of the nucleosomes within the replication origin sequence is the cause of this reduction by limiting the association of the Origin Recognition Complex (ORC). The authors then test whether the chromatin templates are competent for origin activation and resulting DNA replication, both in an extract that provides the downstream components and then in a fully-purified system. The results in both systems further emphasize that a specific chromatin configuration is required for the most-efficient replication activity, highlighting a critical window for establishing this chromatin context before/during the helicase priming step by the Dbf4-dependent kinase (DDK), which is required for multiple downstream steps leading to initiation of replication.

The work in this manuscript provides an important step in dissecting how chromatin influences DNA replication initiation. The strengths lie in the ability to profile, in a controlled manner, multiple CREs to demonstrate that distinct chromatin organization affects specific substeps in the mechanism. The manuscript could be significantly improved by the addition of a small set of experiments and data. Most notably, incorporating experiments using an ORC1 mutant lacking the bromo-adjacent homology (BAH) domain, previously demonstrated to bind nucleosomes and influence replication initiation in vivo.

Major Comments:

1) The manuscript does not mention one of the main interaction points between the chromatin and the initiation machinery: the BAH domain of ORC1 that can bind nucleosomes. This domain has been shown in yeast to affect certain initiation events and in humans this domain has been implicated in both reading an origin "histone code" and potentially in Meier-Gorlin Syndrome. A mutant version of ORC could be produced with the BAH removed and the assays in this manuscript could be repeated. If the domain is not important for any activities described in the paper, this may suggest that post-translational modifications of the histones may be required for BAH-specific activity. Alternatively, the BAH domain may be necessary for ORC, when pre-bound to naked DNA, to resist the conversion of the chromatin organization by ISW2 to a non-permissive state.

2) SWI/SNF and RSC CREs generate chromatin organization that is not permissive to origin activation, yet it is not clear why this occurs. It is also not clear why subsequent treatment with Isw1a rescues activation. The nucleosome occupancy patterns in Figure 5—figure supplement 2 do not shed light on this. One possibility is that there are altered nucleosomes around the origin that are formed by SWI/SNF or RSC that are not adequately profiled by the MNase-Seq, which is described as sequencing only mono-nucleosomal fragments less than 160 bp. Alternatively, is there a possibility that the intensity of the nucleosomes is not easily comparable across different templates, due to internal normalization? The profiles of the whole 3.8 kb template for each remodeling condition may help to address this, or using some independent method to fragment the DNA and sequencing could address% occupancy. Addressing these points would at the very least rule out certain caveats.

Reviewer #2:

Main scientific findings:

The manuscript addresses the issue of how chromatin structure influences/ regulates replication initiation at two steps: (i) origin licensing by orc binding to origins and (ii) origin activation by recruitment and activation of the MCM2-7 helicase. The main conclusions are, derived from in vitro reconstitution of chromatin in a template DNA containing an ars (ori) using different chromatin remodeling complexes, are (i) chromatin structure enhances the specificity of orc binding to ori, (ii) although remodeling complexes inhibit MCM activity when added during the reaction, pre-incubation with remodelers to the template before the addition of MCM and other proteins of the pre-initiation complex rescued replication. Thus, the manuscript demonstrates, perhaps as expected, that nucleosomal landscape modulates replication just as it does of other DNA transactions such as transcription, DNA repair, replicative life span control.

Significance:

The authors get credit for biochemically addressing the relationship between chromatin structure and replication initiation. There are two other papers in Molecular Cell on this subject from Diffley's group from UK that has carried the work into chromatin structure and its regulatory role in replication initiation further. This reviewer is of the opinion that the work by the other group should not in any way diminish the impact of the present manuscript from the point of view of publication priority.

Critique:

The work presented in this manuscript is well done. However, the evidence for specific ori-dependent initiation in the opinion of this reviewer does not meet the gold standards established by Kornberg and others in oriC, plasmid replication reconstitution and SV40 replication. Without 2D gel analyses or EM, it was difficult to assess to what extent initiation fidelity was enhanced by the chromatin remodeling factors in effectively limiting initiation to the ars sequences. After all, in vivo no initiation occurs under normal conditions without the presence of an ars.

A second question is how does chromatin remodeling enhance the specificity of licensing and activation of the origins? Is it just by repositioning the nucleosomes around the region of the ars and thereby restricting orc binding to the origins and preventing its physiologically unwarranted binding to the non-ars sequences or is it more intricate? Diffley's group finding that FACT is involved in the process suggests a more elaborate reaction involving perhaps transcription. It was reported by Marahrens and Stillman that one essential element of the ars directs transcription and is needed for ars activity.

In summary, this is a good paper that explores the role of chromatin in replication and therefore has some novelty. As noted above, the specificity of initiation seen in vivo has not been duplicated in vitro and therefore any conclusions from the chromatin reconstitution work should be cautiously interpreted.

Reviewer #3:

The manuscript by Azmi et al. presents the in vitro reconstitution of DNA replication on chromatinized templates, pre-treated with different chromatin remodeling enzymes (CREs). The authors show how treatment with different CREs lead to differential effects to ORC binding or replication efficiency.

Although it is important to understand how nucleosomes affect DNA replication, I am skeptical about the impact of the data, as they are presented here. In my opinion, the manuscript represents a collection of observations, with insufficient mechanistic insight to warrant publication in *eLife*.

The conclusion that encroachment of the origin inhibits helicase loading is valid. However, the remaining conclusions are not supported by sufficiently controlled mechanistic insight. In particular, the chromatin templates (after remodeling) are insufficiently characterized to allow conclusions on the role of CREs and the nature of the nucleosomes at the origin. The CREs preparation should be tested in a bona fide CRE assay to ensure that they are all "equally" trustworthy/active.

1) The data presented seems to have more to do with the activities of the CREs than with replication. A gold standard CRE assay to validate the activity of preparations should be shown. Some introduction into the mechanistic differences between the different CREs (sliding, exchanging, etc.) have to be included both in the Introduction and in the Discussion.

2) What happens if SWI/SNF or RSC are added to an ISWI1a preparation? Do they overrule ISWI1a? In other words, once you have a good template, can you mess it up (after loading)? To have an impactful conclusion from these studies, a more controlled analysis of the cross-talk between CREs is required, as all CREs are present in the nucleus.

3) It is unclear how the chromatinized plasmid was prepared. In some part of the text, and for example in Figure 1—figure supplement 2, it is stated that the octamer:DNA ratio was 1:1.3, yet from the method section it appears that this is not accurate as the octamer/Nap1 complexes are in large excess to the DNA (180 nM octamer to 120 fmol 3.8kb DNA). This needs to be clarified.

4) Why do you need to purify away the Nap1 and CREs? This is not discussed, but I would argue that you may need a histone binding protein to avoid that the evicted histones precipitate onto DNA again. How much of each CRE remains bound is a critical point.

5) This is linked to point 2. In Figure 1—figure supplement 3, are aCPB and aFLAG on the same membrane? If not, an internal reference should be used to validate the wash of the CREs. The homogenous removal of the CREs is essential in interpreting the results.

6) I am confused by some results/statements:

- The results of the MNAse in presence of SWI/SNF and RSC (Figure 2) shows that the "nucleosomes" in those samples are not stable, yet you could perform the MNase-seq (Figure 5—figure supplement 2) and you could get a 147 bp protection in Figure 2—figure supplement 2. What is the difference between these experiments?

– Related to this, the statement “Of the seven CREs we tested, only SWI/SNF and RSC did not establish regularly-spaced nucleosomes”, as RSC and SWI/SNF do not affect spacing, there are just no stable nucleosomes in those samples.

– The statement “These similarities suggests that local nucleosome positioning and/or states influence replication timing”: nucleosome positioning is not an issue with RSC and SWI/SNF, so I would rephrase this and remove the position component. I also think more controls are needed to make this statement.

7) I am left with the question whether CREs are essential. Can you just add histones and Nap1 to make nucleosomes? How would this "chromatin template" behave in your assays? Would it look like ISWI2/Chd2? This has not been shown, but it may be a good control to strengthen the relevance of CREs in the process.

---

## [Author Response]

*Essential revisions:*

*1) They demonstrate the activity of each CRE in an ATPase assay or on a mononucleosomal template.*

We have included a new figure (Figure 1—figure supplement 3) showing ATPase activity of each CRE. Based on these measurements, we used equal ATPase activity for each CRE for all the assays.

*2) They test whether the presence of a histone chaperone has an effect on the replication assays, where histones are presumably displaced.*

We have included a new figure showing the effect of adding Nap1 or FACT to our assays. In each case we do not see stimulation of replication (see Figure 7). This is likely because other factors are limiting the elongation phase of the assay (e.g. Mrc1, see (Yeeles et al., 2017)). We note that we do not intend this study to address the elongation stage of the reaction as other previous papers have (Devbhandari et al., 2016; Kurat et al., 2016). Instead, we are using the replication phase of our assays as a simple readout of the number of replication forks that are established. That this is the case, is reflected in the close correlation that we observe between the extent of CMG complex formation and the amount of replication products throughout our assays. Although we will include this data in the final manuscript if the reviewers feel strongly, we do not think it enhances our conclusions about the effect of chromatin on replication initiation events and may confuse the readers as to the goals of our study.

Author response image 1.**DOI:**
http://dx.doi.org/10.7554/eLife.22512.028

3) They provide a better description of how the chromatin was prepared.

The method we employed to prepare nucleosome DNA templates was previously published from Dr. Carl Wu’s lab (Mizuguchi et al., 2012) and optimization was carried out with close communication with Dr. Mizuguchi. The final ratio of DNA: octamer used in our assays is 1:1.3 by mass (296ng of DNA and 385ng of octamers), which is now indicated in the description of the template preparation (subsection “SWI/SNF and RSC templates reduce CMG formation”).

*4) They discuss the caveat of why the specificity of initiation seen* in vivo *has not been duplicated using their* in vitro *system.*

In contrast to the reviewer’s suggestion, our helicase loading and initiation reactions exhibit sequence specificity that is consistent with that observed in vivo. Contrary to studies from the Diffley and Remus labs (Devbhandari et al., 2016; Gros et al., 2014; Kurat et al., 2016), the conditions that we use for our helicase loading and replication initiation assays require an intact *ARS1* sequences – in particular the highly-conserved ARS consensus sequence that binds ORC (Heller et al., 2011; Ticau et al., 2015). We show that this is the case for helicase-loading reactions with and without nucleosomes in Figure 1, bottom row (all our reactions are normally performed at 300 mM K-glutamate). Consistent with the requirement of helicase loading for replication initiation, our replication reactions are also strongly dependent on an intact origin. We have added an additional experiment illustrating this dependence (Figure 3—figure supplement 1) which we describe in the subsection “Local nucleosomes impact replication events downstream of helicase loading”.

The only exception to this origin-dependence in our manuscript is that we deliberately promoted non-specific helicase loading on naked DNA by using lower-salt conditions (Devbhandari et al., 2016; Kurat et al., 2016; Remus et al., 2009) for Figure 1 mM K-glutamate, top row). We did this to ask whether nucleosomes would enhance origin specificity under conditions in which origin selection is less specific. Indeed, we found that nucleosomes make helicase loading origin-specific even when naked DNA loading is origin-non-specific.

*Other Comments:*

*1) The manuscript does not mention one of the main interaction points between the chromatin and the initiation machinery: the BAH domain of ORC1 that can bind nucleosomes. This domain has been shown in yeast to affect certain initiation events and in humans this domain has been implicated in both reading an origin "histone code" and potentially in Meier-Gorlin Syndrome. A mutant version of ORC could be produced with the BAH removed and the assays in this manuscript could be repeated. If the domain is not important for any activities described in the paper, this may suggest that post-translational modifications of the histones may be required for BAH-specific activity. Alternatively, the BAH domain may be necessary for ORC, when pre-bound to naked DNA, to resist the conversion of the chromatin organization by ISW2 to a non-permissive state.*

To address the role of the Orc1 BAH domain, we have produced ORC lacking the BAH domain (ORC∆BAH) and studied its effect on helicase loading using nucleosomes assembled with several different CREs (new Figure 2—figure supplement 2). Our results indicate that ORC∆BAH fully substitutes for ORC in these conditions. In particular, we did not observe further decreases in helicase loading for nucleosome assembled by ISW2. These findings suggest that interactions between the Orc1 BAH domain and local nucleosomes are not required for helicase loading at *ARS1*. We note that previous studies indicate that deletion of the BAH domain in vivo has only a limited effect on *ARS1* function (Müller et al., 2010). Although it would be interesting to explore the function of the BAH domains at other origins, this is beyond the scope of the current study. We agree with the reviewer that these findings may also indicate that the effects of the BAH domain require modified versions of histones not present in our current assays. We describe these new findings in subsection “Local nucleosomes reduce helicase loading by inhibiting ORC DNA binding” as well as in the Discussion section.

*2) SWI/SNF and RSC CREs generate chromatin organization that is not permissive to origin activation, yet it is not clear why this occurs. It is also not clear why subsequent treatment with Isw1a rescues activation. The nucleosome occupancy patterns in Figure 5—figure supplement 2 do not shed light on this. One possibility is that there are altered nucleosomes around the origin that are formed by SWI/SNF or RSC that are not adequately profiled by the MNase-Seq, which is described as sequencing only mono-nucleosomal fragments less than 160 bp. Alternatively, is there a possibility that the intensity of the nucleosomes is not easily comparable across different templates, due to internal normalization? The profiles of the whole 3.8 kb template for each remodeling condition may help to address this, or using some independent method to fragment the DNA and sequencing could address percent occupancy. Addressing these points would at the very least rule out certain caveats.*

We agree with the reviewer that MNase-seq may not fully reveal the differences between the origin-proximal nucleosomes for the ISW1a, SWI/SNF or RSC templates. Unfortunately, our original experimental procedure eliminated smaller fragments so we cannot explore this data to identify sub-nucleosomal fragments that might impact our findings. On the other hand, it is not clear how addition of ISW1a would remove or remodel subnucleosomal particles to eliminate this issue. With regard to comparing nucleosome occupancy, we see similar amounts of nucleosomes associated with the different templates (Figure 2—figure supplement 1) and the nucleosome occupancies that we observe in the MNase-Seq experiments are similar between ISW1a, RSC and SWI/SNF (Figure 5—figure supplement 2). We agree that the current data does not provide a clear mechanistic understanding and there is much to be done to address these observations at a detailed mechanistic level. Nevertheless, our data clearly indicates that these differences observed between the different nucleosomal DNA templates is the result of different CREs establishing distinct nucleosomal states.

*3) Why do you need to purify away the Nap1 and CREs? This is not discussed, but I would argue that you may need a histone binding protein to avoid that the evicted histones precipitate onto DNA again. How much of each CRE remains bound is a critical issue.*

We choose to remove the CRE from templates primarily to prevent additional nucleosome assembly and remodeling during the assays. Our intent was to establish different chromatin states and ask how DNA replication initiation responds to them. Nap1 gets removed as a consequence of removing CREs, since removal of CRE requires a stringent wash. Additionally, chaperons are primarily involved in the replication elongation step (e.g. replication-coupled nucleosome removal and assembly (Burgess and Zhang, 2013)). In this manuscript, we focus on replication initiation step (e.g. we have not included all the factors required for lagging-strand synthesis in our assays) and thus we did not focus our efforts in understanding how would chaperones influence replication elongation. Nonetheless, we have performed a replication assay to examine whether Nap1 or FACT influence replication initiation in our assay and found no dramatic increase in replication (see major comments, point 2, above).

We address the amount of each CRE that remains bound to the template (Figure 1—figure supplement 1). In each case, little of no detectable CRE remains. We note that for the case of the RSC and SWI/SNF templates, the effects on replication cannot be due to residual association of these enzymes as adding these enzymes back to the reactions does not restore full levels of replication products (Figure 5 and Figure 5—figure supplement 1). We only observe rescue of the reduced replication capacities of these templates when we add back different CREs (Figure 5 and Figure 5—figure supplement 1) and only when they are added before the helicase-activating proteins (Figure 5).

*4) In Figure 1—figure supplement 3, are anti-CPB and anti-FLAG on the same membrane? If not, an internal reference should be used to validate the wash of the CREs. The homogenous removal of the CREs is essential in interpreting the results.*

In Figure 1—figure supplement 2, the anti-CPB and anti-FLAG immunoblots were performed on the same membrane. After the anti-CPB immunoblot was performed, the membranes were stripped and re-probed for the FLAG-epitope.

*5) Clarify these statements:- "The results of the MNAse in presence of SWI/SNF and RSC (Figure 2) shows that the "nucleosomes" in those samples are not stable". Could you perform the MNase-seq (Figure 5—figure supplement 2) and get a 147 bp protection in Figure 2—figure supplement 2? What is the difference between these experiments?*

The reviewer is correct that we used the incorrect terminology here. What we should have stated is that the nucleosomes are not uniformly spaced. We have revised the text accordingly (subsection “CRE-specific restoration of replication and CMG formation to RSC and SWI/SNF templates“).

The difference in the experiment shown in Figure 2 and those shown for Figure 2—figure supplement 1 and Figure 5—figure supplement 2 is the extent of MNase digestion. Under low levels of digestion, (Figure 2) we observe a relatively random assortment of fragments that indicate that the assembled nucleosomes are not precisely spaced relative to one another. When we add higher amounts of MNase, this random pattern of cleavage between nucleosomes collapses into similar amounts of protected ~147bp DNA for the RSC and SWI/SNF templates as we see for templates that show more uniformly spaced nucleosomes (Figure 2—figure supplement 1). We also have performed MNase-seq with SWI/SNF and RSC templates and observe 147 bp protected fragments as shown in Figure 5—figure supplement 2.

*6) Figure 1 should include a Western blot for ORC as well.*

Figure 1 is a helicase loading reaction that is washed with high-salt. ORC is removed under these conditions and, therefore, would not be detected. We note that the key question in Figure 1 is whether Mcm2-7 has been loaded (the purpose of the high-salt wash is to remove any incompletely loaded Mcm2-7 – a standard method for the field). We also note that we include an ORC blot in Figure 1, where a low-salt wash that does not remove ORC is performed and we see equivalent ORC binding with and without nucleosomes.

*7) Figure 2 would be improved by showing ORC occupancy in all CRE contexts.*

Mcm2-7 loading requires ORC. Given that we did not see defects in helicase loading for the remaining CREs we do not think that including them in Figure 2 will enhance the conclusion of the manuscript. We chose the subset shown to focus on addressing the defects in helicase loading observed for the Isw2 and Chd1 templates. We do show that there are not significant differences for two templates that exhibit the same helicase loading observed for naked DNA (Figure 2) and showing additional ORC blots would not help to explain the defects in loading for Chd1 and Isw2 templates.

*8) The size range of all replication products run on denaturing agarose gels should be indicated with marker indicator.*

We have included this information for each figure.

*9) Dissection of the steps after DDK treatment, using purified components such as Cdc45, would make this manuscript even more impactful.*

We agree that further determination of the precise step impacted by different nucleosomal template in CMG formation would be of interest. Nevertheless, the critical issue is whether the CMG is formed or not and we have clearly documented that it is at this step that nucleosomes assembled in the presence of RSC and SWI/SNF are impacting initiation. Whether this occurs at the Cdc45 or GINS recruiting step will be one of several future directions for these studies.

*10) Including a mutant MCM2 lacking the histone binding domain would increase the impact of the manuscript, similar to the ORC BAH mutant.*

We had done this experiment previously and mentioned it in the Discussion of the previous version of the paper because we did not observe a defect for this mutant. We have now included this experiment comparing helicase loading and replication initiation using Mcm2-7 with and without the histone-binding domain in the Results section (subsection “CRE-specific restoration of replication and CMG formation to RSC and SWI/SNF templates” paragraph three; Figure 5—figure supplement 3).

*11) Including the Abf1 protein, which binds the B3 element of ARS1, would increase the impact of the manuscript. Previously it was proposed that Abf1 helps establish a nucleosome free region for MCM loading.*

To address a possible role for Abf1, we examined whether pre-binding of Abf1 to DNA prior to ISW2 nucleosome assembly rescued the defects in helicase loading observed (Figure 2—figure supplement 2). Unlike ORC pre-binding to the DNA, we found that Abf1 is not sufficient to rescue the defect in helicase loading in the presence of ISW2. Additionally, when Abf1 is included during ISW1a nucleosome assembly, we do not see an increase or decrease in the ability of ISW1a templates to load Mcm2-7. Although prior genetic studies show that Abf1 binding enhances replication initiation (Marahrens and Stillman, 1992), this is in the context of a complete chromosome with the adjacent *TRP1* gene being transcribed into the *ARS1* origin. We suspect that Abf1 may become more important under these conditions.

*We have included the full reviews at the bottom of this letter, to further clarify these requests.*

*Reviewer #2:*

[…]

*A second question is how does chromatin remodeling enhance the specificity of licensing and activation of the origins? Is it just by repositioning the nucleosomes around the region of the ars and thereby restricting orc binding to the origins and preventing its physiologically unwarranted binding to the non-ars sequences or is it more intricate? Diffley's group finding that FACT is involved in the process suggests a more elaborate reaction involving perhaps transcription. It was reported by Marahrens and Stillman that one essential element of the ars directs transcription and is needed for ars activity.*

We believe that the effect of nucleosomes on helicase loading is simply due to nucleosome occlusion and this conclusion is supported by our studies of local nucleosome positioning (Figure 2). The role of nucleosomes in the helicase activation process remains less clear. We provide strong evidence that the effects that we see are due to nucleosomes since they are readily modulated by CREs. Because our reactions do not include any components of the transcription apparatus (they are all performed with purified replication proteins – see (Looke et al., 2017) for gels of the proteins used) any effect of Abf1 or other added proteins on transcription would not be observed in our assays. The same is true for the Diffley lab experiment addressing the role of FACT in replication (Kurat et al., 2016). Instead, FACT is likely to be functioning in the Diffley studies to remove nucleosomes in front of the replication fork (as has been described for FACT’s role in transcription of chromatin templates (Belotserkovskaya et al., 2004)). Indeed, one of the FACT subunits was originally discovered due to its association with DNA Pol α (Miles and Formosa, 1992; Wittmeyer and Formosa, 1997).

Finally, the factor that the reviewer is referring to in the Marahrens and Stillman paper is Abf1(Marahrens and Stillman, 1992), whose function we now address (Figure 2—figure supplement 2). Although this protein can act as a transcription factor in some situations, there is no evidence that it does so at *ARS1*. Instead, Abf1 is known to be involved in the establishment of a positioned nucleosome adjacent to *ARS1* (Lipford and Bell, 2001). in vivo this may be particularly important since the *TRP1* gene transcribes into the *ARS1* region on the Abf1-side of the origin DNA.

*Reviewer #3:*

[…]

*1) The data presented seems to have more to do with the activities of the CREs than with replication. A gold standard CRE assay to validate the activity of preparations should be shown. Some introduction into the mechanistic differences between the different CREs (sliding, exchanging, etc.) have to be included both in the Introduction and in the Discussion.*

The first issue is addressed above: major comments, point 1. We have also added additional information about the mechanistic differences between the CREs used into the Introduction (paragraph four) and Discussion section.

*2) What happens if SWI/SNF or RSC are added to an ISWI1a preparation? Do they overrule ISWI1a? In other words, once you have a good template, can you mess it up (after loading)? To have an impactful conclusion from these studies, a more controlled analysis of the cross-talk between CREs is required, as all CREs are present in the nucleus.*

We have done this experiment and found that SWI/SNF or RSC cannot reduce the efficiency of a ISW1a template. This suggests that once a chromatin state that is active for CMG formation is formed, it is not easily removed. This could be due to the different activities of the CREs or because there are positive interactions formed (e.g. between Mcm2-7 and nucleosomes) that are difficult for RSC or SWI/SNF to eliminate. We have added this data to the Results section(subsection “CRE-specific restoration of replication and CMG formation to RSC and SWI/SNF templates”; Figure 5—figure supplement 1) and discuss the implications of the findings (subsection “CREs show different capacities to establish replication-competent chromatin states.”).

*3) It is unclear how the chromatinized plasmid was prepared. In some part of the text, and for example in Figure 1—figure supplement 2, it is stated that the octamer:DNA ratio was 1:1.3, yet from the method section it appears that this is not accurate as the octamer/Nap1 complexes are in large excess to the DNA (180 nM octamer to 120 fmol 3.8kb DNA). This needs to be clarified.*

Addressed above, major comment 3. We note that the relative molarity of the plasmid DNA and the octamer is not the correct comparison, as different lengths of DNA would provide different total amounts of DNA in the reaction for the same molarity. Instead, one has to determine the relative total amounts of DNA and octamer (in our case this is a 1:1.3 ratio or 296ng of DNA and 385ng of octamers in a typical reaction).

*4) Why do you need to purify away the Nap1 and CREs? This is not discussed, but I would argue that you may need a histone binding protein to avoid that the evicted histones precipitate onto DNA again. How much of each CRE remains bound is a critical point.*

Addressed above: other comments, point 3.

*5) This is linked to point 2. In Figure 1—figure supplement 3, are aCPB and aFLAG on the same membrane? If not, an internal reference should be used to validate the wash of the CREs. The homogenous removal of the CREs is essential in interpreting the results.*

Addressed above: other comments, point 4.

*6) I am confused by some results/statements:*

*- The results of the MNAse in presence of SWI/SNF and RSC (Figure 2) shows that the "nucleosomes" in those samples are not stable, yet you could perform the MNase-seq (Figure 5—figure supplement 2) and you could get a 147 bp protection in Figure 2—figure supplement 2. What is the difference between these experiments?*

Addressed above: other comments, point 5.

*– Related to this, the statement “Of the seven CREs we tested, only SWI/SNF and RSC did not establish regularly-spaced nucleosomes”, as RSC and SWI/SNF do not affect spacing, there are just no stable nucleosomes in those samples.*

We disagree that the inability of RSC and SWI/SNF to space nucleosomes indicates that there are no stable nucleosomes on these templates. Both the equivalent levels of 147 bp protected DNA (Figure 2—figure supplement 2), the similar patterns of origin proximal nucleosomes between ISW1a and SWI/SNF (Figure 5—figure supplement 2), and the equivalent levels of bound histones (e.g. Figure 3, Figure 4) all indicate that the RSC and SWI/SNF templates include stably-associated nucleosomes.

*– The statement “These similarities suggests that local nucleosome positioning and/or states influence replication timing”: nucleosome positioning is not an issue with RSC and SWI/SNF, so I would rephrase this and remove the position component. I also think more controls are needed to make this statement.*

We agree that local nucleosome states do not appear to be an issue with SWI/SNF vs. ISW1a, however, there are significant differences for RSC and ISW1a as well as for RSC vs ISW1a treated RSC templates. Thus, we think that including changes in nucleosome positioning in this statement is appropriate. It is not clear what other controls the reviewer is referring to. We note, however, that the statement at issue is only suggesting a possibility raised by our data, not making a conclusion.

*7) I am left with the question whether CREs are essential. Can you just add histones and Nap1 to make nucleosomes? How would this "chromatin template" behave in your assays? Would it look like ISWI2/Chd2? This has not been shown, but it may be a good control to strengthen the relevance of CREs in the process.*

We have assembled nucleosomes with only and histones and Nap1 and the MNase pattern looks similar to that of RSC or SWI/SNF, however, helicase loading and replication initiation is similar to that of ISW1a or naked DNA templates. Not surprisingly, when analyzed by limited MNase digestion, these templates show a similar lack of uniformly-spaced nucleosomes as we observe for RSC and SWI/SNF templates. Thus, the different replication capacities we observe for the RSC and SWI/SNF templates requires the action of these enzymes and is not simply a consequence of non-uniformly spaced nucleosomes. We have now included this experiment as Figure 4—figure supplement 2 and modified the text accordingly (subsection “RSC and SWI/SNF templates impede CMG complex formation” and “CRE-specific restoration of replication and CMG formation to RSC and SWI/SNF templates”).